# Maximizing Incremental Information Entropy for Contrastive Learning

**Jiansong Zhang**[1,3], **Zhuoqin Yang**[1,3], **Xu Wu**[1], **Xiaoling Luo**[1], **Peizhong Liu**[2], **Linlin Shen**[1,3,4*]

[1] School of Computer Science and Software Engineering, Shenzhen University, Shenzhen, China
[2] School of Engineering, Huaqiao University, Quanzhou, China
[3] Computer Vision Institute, School of Artificial Intelligence, Shenzhen University, Shenzhen, China
[4] Guangdong Provincial Key Laboratory of Intelligent Information Processing, Shenzhen University, Shenzhen, China

## Abstract

Contrastive learning has achieved remarkable success in self-supervised representation learning, often guided by information-theoretic objectives such as mutual information maximization. Motivated by the limitations of static augmentations and rigid invariance constraints, we propose **IE-CL** (Incremental-Entropy Contrastive Learning), a framework that explicitly optimizes the entropy gain between augmented views while preserving semantic consistency. Our theoretical framework reframes the challenge by identifying the encoder as an information bottleneck and proposes a joint optimization of two components: a learnable transformation for entropy generation and an encoder regularizer for its preservation. Experiments on CIFAR-10/100, STL-10, and ImageNet demonstrate that IE-CL consistently improves performance under small-batch settings. Moreover, our core modules can be seamlessly integrated into existing frameworks. This work bridges theoretical principles and practice, offering a new perspective in contrastive learning.

## 1 Introduction

Self-supervised contrastive learning has emerged as a cornerstone paradigm for representation learning, enabling models to extract rich semantic features without explicit labels. At its core, contrastive learning constructs a latent space where semantically similar samples converge while dissimilar ones diverge Wang & Isola (2020); Le-Khac et al. (2020). Despite rapid advances in this field, a fundamental tension persists between theoretical understanding and practical implementation. While recent works have decomposed contrastive objectives into *alignment* and *uniformity* principles Wang & Isola (2020); Zhang et al. (2024), they offer limited insight into the *dynamic information landscape* that unfolds during the learning process.

Current contrastive frameworks such as SimCLR Chen et al. (2020a), MoCo He et al. (2020), and their variants rely heavily on static, human-engineered augmentations and large batch sampling to enforce invariance and representational diversity. These approaches impose rigid constraints on the learning dynamics: augmentations must balance semantic preservation with transformational complexity, while batch scaling faces inevitable hardware limitations. Despite substantial engineering efforts to refine augmentation strategies Chen & He (2021); Chen et al. (2021); Tian et al. (2020a), these methods fundamentally lack a principled mechanism for adaptively expanding the representational capacity of each instance while maintaining semantic coherence.

We address this limitation by reconceptualizing contrastive learning through the lens of *incremental information entropy*, a novel framework that quantifies the expansion of representation space during learning. Inspired by information-theoretic objectives Hjelm et al. (2019); Bardes et al. (2022), we focus on how additional controllable uncertainty is gained between augmented views to strengthen learning. Our key insight is that optimal contrastive learning could maximize the conditional entropy gain between positive views while preserving their mutual information. The effectiveness of maximizing this incremental entropy is contingent on its preservation through the deep encoder,

---

*Corresponding author: `llshen@szu.edu.cn`

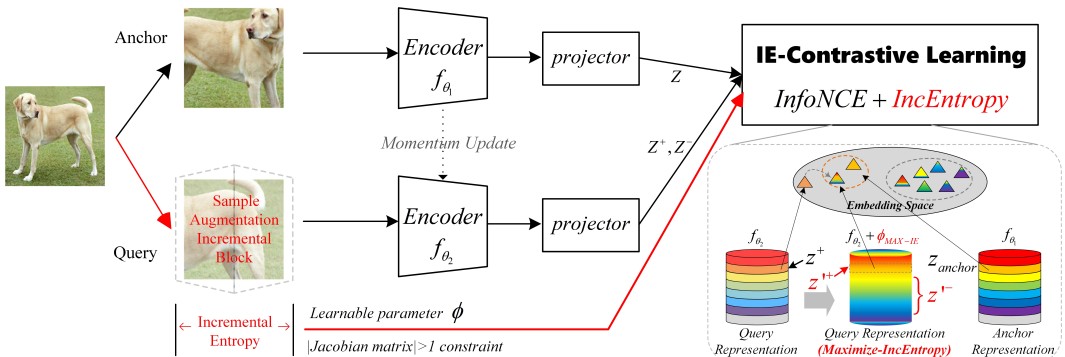

Figure 1: **Overview of the proposed IE-CL.** We define incremental entropy as the absolute change in entropy induced by classical contrastive augmentations (see Definition 3.2). To optimize the contrastive learning process, we propose the **S**ample **A**ugmentation **I**ncremental **B**lock (**SAIB**), a learnable module that ensures the local Jacobian determinant > 1. By incorporating sample-level incremental entropy into contrastive optimization, we establish a principled framework that improves the effectiveness of self-supervised representation learning.

which often acts as an information bottleneck. We therefore propose a framework that jointly optimizes two synergistic components: a learnable transformation for *entropy generation* and an explicit regularization of the encoder for *entropy preservation*. This principled approach highlights the overlooked trade-off between semantic invariance and representational expressivity.

Based on this theoretical insight, we introduce IE-CL (**I**ncremental-**E**ntropy **C**ontrastive **L**earning), a framework that explicitly optimizes for controlled entropy gain. To achieve this, we design a learnable nonlinear transformation module (SAIB) that adaptively expands each sample's local representation manifold by guaranteeing a strictly positive Jacobian determinant. Crucially, to ensure this generated entropy is not lost during encoding, this module is paired with an explicit encoder regularization mechanism that encourages information preservation. These components work in concert with a Kullback-Leibler divergence constraint to balance entropic expansion against semantic consistency. IE-CL operates efficiently under small batch sizes (e.g., 256), enabling broader applicability without the hardware burden of large-batch training.

The contributions of this work can be summarised as: (1) We propose a new theoretical framework for contrastive learning that identifies the deep encoder as an information bottleneck. We posit that effective representation learning requires jointly optimizing for both **entropy generation** at the input and **entropy preservation** during encoding. (2) Based on this framework, we design a novel model, IE-CL, featuring two key components: a learnable transformation (SAIB) to generate rich input-level entropy, and an encoder regularizer (e.g., Spectral Normalisation) to ensure its faithful propagation. (3) We provide a detailed empirical analysis demonstrating IE-CL's effectiveness, particularly in small-batch settings. We also show that our core module can enhance other self-supervised models in a plug-and-play manner.

Our work bridges the gap between information-theoretic principles and practical contrastive learning, offering a more complete theoretical understanding and algorithmic innovations that significantly advance the field of self-supervised representation learning.

## 2 RELATED WORK

**Self-supervised Paradigm** Self-supervised learning has emerged as a prominent paradigm for feature extraction without reliance on labeled dataLiu et al. (2022); Wang et al. (2023); Yang et al. (2024). A central research focus has been the development of effective encoding frameworks that facilitate rich representation learning in the absence of supervision. Notable approaches include contrastive learningChen et al. (2020b;c); Chen & He (2021); Chen et al. (2021); Caron et al. (2021); Oquab et al. (2024); Wu et al. (2023), masked autoencodersZhou et al. (2022b); Xie et al. (2022); Wei et al. (2022); Chen et al. (2024), and advances in loss function designErmolov et al. (2021); Zbontar et al. (2021); Tian et al. (2020b); Ozsoy et al. (2022); Bardes et al. (2022). Among these, contrastive

learning has become a dominant paradigm due to its ability to extract rich features through well-designed pretext tasks within a dual-encoder frameworkGarrido et al. (2023).It has frequently served as a benchmark for evaluating self-supervised learning methods. Recently, the emergence of masked pretext tasks has opened new avenues for learning representations in a label-free setting. Works such as He et al. (2022) and Bao et al. (2022) creatively adapted masking strategies from NLP to vision, enabling image reconstruction from masked tokens using spatial priors and positional embeddings. Following this, Jinghao Zhou et.alZhou et al. (2022a) further abstracted feature representations in image self-supervised learning using a knowledge distillation-based masking learning strategy, also demonstrating the effectiveness of masking strategies in dual-track self-supervised frameworks like contrastive learning. Concurrently, the realm of non-masking pretext tasksMo et al. (2023); Huang et al. (2022); Oinar et al. (2023) in self-supervised learning has witnessed numerous novel contributions. Notably, Tong et.alTong et al. (2023) employed an extremely high number of patches as a self-supervised signal, proposing a self-supervised learning framework requiring only one epoch. The remarkable success of these works is largely attributable to researchers' deepening understanding of data processing methods in self-supervised learning.

**Contrastive Learning Theory** The empirical success of contrastive learning has spurred extensive theoretical investigations. Early work focused on analyzing the mathematical foundations of contrastive loss. Saunshi et al. Saunshi et al. (2019) were among the first to show that contrastive learning can produce linearly separable representations under certain conditions. Wang and Isola Wang & Isola (2020) decomposed the InfoNCE loss into two interpretable terms—alignment and uniformity—where alignment promotes similarity between positive pairs and uniformity mitigates feature collapse. This formulation offered a unified lens for understanding contrastive learning and inspired connections to broader information-theoretic frameworks, such as mutual information maximization Tian (2022) and noise contrastive estimation Hu et al. (2023). From a spectral graph theory viewpoint, Chen et al. HaoChen et al. (2021) revealed that contrastive learning implicitly learns the Laplacian of the data graph, showing equivalence to spectral clustering objectives. This was later extended to dynamic graphs Shen et al. (2022) and connected to kernel methods Wang et al. (2022). Tan et al. Tan et al. (2024) introduced $\alpha$-order mutual information to unify contrastive and non-contrastive losses, bridging matrix-based contrastive methods (e.g., Barlow Twins Zbontar et al. (2021), VICReg Bardes et al. (2022)) with classical dimensionality reduction techniques such as ISOMAP. Beyond spectral perspectives, Zimmermann et al. Zimmermann et al. (2021) proposed a probabilistic interpretation, viewing contrastive learning as reverse engineering the data generation process under the assumption of a uniform latent prior. This aligns with the framework of noise contrastive estimation Gutmann & Hyvärinen (2010) and sheds light on its generalization behavior. Lee et al. Lee et al. (2021) further established a link between contrastive loss and the variational lower bound of the data likelihood using latent variable models. As non-contrastive approaches such as BYOL Grill et al. (2020) and Barlow Twins Zbontar et al. (2021) gained popularity, recent efforts have focused on theoretically characterizing the distinction between contrastive and non-contrastive paradigms Zhang et al. (2024).

## 3 METHOD

### 3.1 INFORMATION ENTROPY IN CONTRASTIVE LEARNING

**Contrastive Learning Objectives** The primary goal of contrastive learning is to optimize the similarity between positive pairs (anchor and query) while repelling negative samples, thereby enabling effective self-supervised representation learning under the assumption of independently and identically distributed (i.i.d.) samples within a mini-batch. For a given batch of original images $B = \{x_i \mid i = 1, 2, \ldots, N\}$, the representation $z_i \in Z_1$ denotes the embedding of image $x_i$, computed via the encoder $f_{\theta_1}$. This embedding typically originates from the anchor branch in a dual-stream contrastive architecture. The representation $z_i^+$ denotes the positive sample, whereas $z_j \in Z_2$(with $j \neq i$) corresponds to negative samples derived from different instances in the batch. These negative and positive representations are encoded by the second branch, $f_{\theta_2}$, and are collectively referred to as the query set. The standard form of the objective can be expressed as:

$$L(\mathbf{Z}_1, \mathbf{Z}_2) = -\frac{1}{N} \sum_{i=1}^{N} \log \frac{\exp(\text{sim}(z_i, z_i^+)/\tau)}{\sum_{j=1}^{N} \exp(\text{sim}(z_i, z_j)/\tau)} \tag{1}$$

where $\text{sim}(z_i, z_j)$ is the similarity function, and $\tau$ is the temperature parameter that controls the sharpness of the probability distribution. Cosine similarity is often employed, defined as:

$$\text{sim}(z_i, z_j) = \frac{z_i^\top z_j}{\|z_i\|\|z_j\|} \tag{2}$$

The optimization objective seeks to minimize the distance between each anchor and its positive counterpart $(z_i, z_i^+)$, while maximizing the separation from all negative samples $z_j \neq z_i^+$, thereby facilitating effective self-supervised learning.

**Mutual Information Theory**   Mutual information provides a principled framework for analyzing self-supervised learning objectives, as discrete probability distributions can be interpreted as samples drawn from an underlying continuous distribution.

**Lemma 3.1** (Equivalence between InfoNCE minimization and mutual information maximization). *Let $Z = f_\theta(X)$ be the embedding of input $X$ and $Z^+$ the corresponding positive sample. Then, based on the Donsker–Varadhan representation, the mutual information satisfies*

$$\min L_{\textit{InfoNCE}} \iff \max I(Z; Z^+), \quad I(Z; Z^+) \geq \log N - L_{\textit{InfoNCE}}.$$

*Proof.* The InfoNCE loss for a positive pair $(z, z^+)$ can be written as

$$L = -\mathbb{E}_{p(z,z^+)}\left[\log \frac{\exp(\text{sim}(z, z^+)/\tau)}{\exp(\text{sim}(z, z^+)/\tau) + \sum_{j=1}^{N-1} \exp(\text{sim}(z, z_j^-)/\tau)}\right]. \tag{3}$$

Using the Donsker–Varadhan representation,

$$I(Z; Z^+) = \sup_T \; \mathbb{E}_{p(z,z^+)}[T(z, z^+)] - \log \mathbb{E}_{p(z)p(z^+)}[e^{T(z,z^+)}]. \tag{4}$$

Choosing $T(z, z^+) = \text{sim}(z, z^+)/\tau$ yields the lower bound

$$I(Z; Z^+) \geq \log N - L_{\text{InfoNCE}}. \tag{5}$$

Thus, minimizing $L_{\text{InfoNCE}}$ is equivalent to maximizing $I(Z; Z^+)$.

## 3.2   Incremental Entropy in Contrastive Learning

It is evident that optimizing the distributions of $Z_1$ and $Z_2$ fundamentally depends on obtaining effective and discriminative feature representations. From an information-theoretic standpoint—abstracting away encoder-specific inductive biases, the learning objective can be intuitively framed as minimizing the conditional entropy $H(Z^+|Z)$ while maximizing the marginal entropy $H(Z^+)$. The incremental entropy is thus defined first from the input side.

**Definition 3.2** (Based on the concept of Shannon Entropy, the change in information entropy of a given sample $X$ after a transformation $g$ is applied, resulting in $X'$, is referred to as the **Incremental Information Entropy**).

$$\Delta H(X) = H(X') - H(X), \quad H(X) = -\sum_i p(x_i) \log p(x_i)$$

The relationship between a transformation and the change in entropy can be precisely quantified. For a linear transformation $g$ represented by a matrix $A$, the incremental information entropy is given by:

$$\Delta H(X) = H(g(X)) - H(X) = \log|\det A|$$

*Proof.* When the transformation $g$ is a linear function, the probability density function of $x$ can be written as:

$$p_X'(x') = p_X\left(A^{-1}(x' - b)\right) \cdot \frac{1}{|\det A|} \tag{6}$$

Replacing $p_X'(x')$ with $H(X')$:

$$H(X') = -\int p'_X(x') \log p'_X(x') \, dx'$$

$$= -\int p_X(A^{-1}(x'-b)) \cdot \frac{1}{|\det A|} \log \left( p_X(A^{-1}(x'-b)) \cdot \frac{1}{|\det A|} \right) dx' \tag{7}$$

Logarithmic term expansion:

$$H(X') = -\int p_X(A^{-1}(x'-b))$$

$$\cdot \frac{1}{|\det A|} \left[ \log p_X(A^{-1}(x'-b)) - \log |\det A| \right] dx' \tag{8}$$

Split into two parts:

$$H(X') = -\int p_X(A^{-1}(x'-b))$$

$$\cdot \frac{1}{|\det A|} \log p_X(A^{-1}(x'-b)) \, dx' + \log |\det A| \tag{9}$$

Perform a permutation on the variable $u = A^{-1}(x'-b)$ with $dx' = |\det A| du$:

$$H(X') = -\int p_X(u) \log p_X(u) \, du + \log |\det A| \tag{10}$$

To wit:

$$H(X') = H(X) + \log |\det A| \tag{11}$$

Incremental information entropy is:

$$\Delta H(X) = H(X') - H(X) = \log |\det A| \tag{12}$$

This relationship makes it clear why standard augmentations have limitations. When the transformation $g$ is a linear isometry (such as rotation, cropping, mirroring, etc.), its matrix representation $A$ has a determinant $|\det A| = 1$, which leads to $\Delta H = 0$. In such cases, these augmentations can enrich sample diversity at the batch-level without altering the instance-level entropy.

However, a critical challenge arises from the nature of deep encoders themselves. In information theory, the Data-Processing Inequality states that post-processing cannot increase information. For differential entropy, this implies that the entropy of a variable's representation $Z = f(X)$ is bounded by the entropy of the original variable $X$. Specifically, for a deterministic function $f$, the change in entropy is governed by:

$$H(f(X)) \leq H(X) + \mathbb{E}_{p(x)}[\log |\det J_f(x)|] \tag{13}$$

where $J_f(x)$ is the Jacobian of the transformation $f$ at $x$. This inequality highlights a crucial issue in representation learning: a deep encoder, acting as the function $f$, can potentially become an information bottleneck, diminishing the entropy of its input. Any diversity generated at the input level is not guaranteed to be preserved in the final representation space. To address this, we introduce the IE-CL framework, a holistic approach that pairs an entropy generation module with an entropy-preserving encoder. We formalize this approach in the following proposition.

**Proposition 3.3** (Principle of Constrained Incremental Entropy Maximization). *Let $X^-$ be a negative sample, $g_\phi$ be a non-linear transformation, and $Z'^- = f_\theta(g_\phi(X^-))$ be the final representation encoded by an encoder $f_\theta$. To robustly increase the representation entropy $H(Z'^-)$, maximizing the input-level incremental entropy $\Delta H(X^-)$ alone is insufficient. A joint condition is required: (1)* **Input Entropy Generation**: *The transformation $g_\phi$ must be optimized to maximize the incremental entropy $\Delta H(X^-)$. (2)* **Encoder Entropy Preservation**: *The encoder $f_\theta$ must be simultaneously constrained to preserve the entropy of its input. Satisfying both conditions provides a principled path toward maximizing the diversity of negative representations for effective contrastive learning.*

*Theoretical Argument.* Our argument is based on the Data-Processing Inequality for differential entropy. Let $X' = g_\phi(X^-)$ be the transformed input to the encoder. The entropy of the final representation, $Z'^- = f_\theta(X')$, is bounded as follows:

$$H(Z'^-) = H(f_\theta(X')) \leq H(X') + \mathbb{E}_{p(x')}[\log |\det J_{f_\theta}(x')|] \tag{14}$$

This inequality reveals the core challenge. The first condition, maximizing $\Delta H(X^-)$, is equivalent to maximizing $H(X')$ since $H(X^-)$ is a constant with respect to the parameters $\phi$ of $g_\phi$. However,

even if $H(X')$ is large, the second term, which depends on the Jacobian of the encoder $f_\theta$, can be a large negative value, effectively nullifying the gains from the first term. This occurs if the encoder acts as a strong information bottleneck, aggressively compressing its input space.

Therefore, to guarantee that a large $H(X')$ induces a correspondingly large $H(Z'^-)$, we introduce the second requirement: constraining the encoder. Specifically, by regularizing $f_\theta$ to be entropy-preserving (e.g., via Lipschitz continuity constraints), we effectively bound the term $\mathbb{E}[\log|\det J_{f_\theta}|]$, thus preventing it from becoming excessively negative. This condition ensures that the entropy injected by $g_\phi$ is faithfully propagated to the final representation space.

Consequently, the joint optimization of an entropy-generating transformation and an entropy-preserving encoder is a necessary and sufficient strategy to robustly increase the final representation entropy $H(Z'^-)$.

### 3.3 MAXIMIZING INCREMENTAL INFORMATION ENTROPY

Based on the framework established in Proposition 3.3, our goal is to co-optimize both the generation of incremental entropy and its preservation through the encoder. While encoder regularization is implemented via standard techniques such as spectral normalization, the core of our contribution lies in the design of a learnable, entropy-generating transformation $g_\phi$. Isometric transformations, as discussed, cannot linearly provide incremental information entropy. To address this, we propose a nonlinear transformation implemented via batch-level pixel-wise operations, explicitly designed to induce positive entropy increments in the query branch.

**Sample Augmentation Incremental Block (SAIB)**  Our objective is to maximize mutual information by minimizing the conditional entropy $H(Z^+ \mid Z)$ on the query side. To inject a semantics-preserving but entropy-expansive transform into the **query branch**[1] we introduce the *SAIB* module, which couples ViT-style positional encoding Dosovitskiy et al. (2021) with a non-linear residual stack. The input $X \in \mathbb{R}^{3 \times H \times W}$ is first patchified into a matrix $P \in \mathbb{R}^{(C\,H/p\,W/p) \times (p^2)}$ (as in ViT, $C = 3$), where the *mini-batch* occupies the channel dimension. A sequence of $1 \times 1 - \mathrm{Conv} \to 3 \times 3 - \mathrm{Conv} \to 1 \times 1 - \mathrm{Conv}$ layers—with channel expansion ratio 2—is wrapped by two skip connections (see Appendix Figure 6). Owing to the channel-expanding residual design, the local Jacobian $A$ of SAIB satisfies $|\det A| > 1$ almost everywhere (Appendix C.1), guaranteeing positive incremental entropy $\Delta H(P) > 0$. After the non-linear block we reshape $P'$ back to the spatial layout and add a troisième skip connection $X' = X + \mathrm{reshape}(P')$.

**KL regularisation to avoid degenerate $g_\phi$.**  Because $g_\phi$ acts only on the query branch, aggressive entropy expansion may lead to distributional drift. We therefore penalise the *Kullback–Leibler divergence*

$$D_{\mathrm{KL}}(p_\phi \parallel q) \;=\; \int p_\phi(\mathbf{z}) \log \frac{p_\phi(\mathbf{z})}{q(\mathbf{z})}\,d\mathbf{z}, \tag{15}$$

where $p_\phi(\mathbf{z}) = p(Z^{-\prime} = \mathbf{z})$ is the SAIB-transformed query distribution and $q(\mathbf{z}) = p(Z = \mathbf{z})$ is the anchor distribution. Assuming $q$ is Gaussian with mean $\mu$ and variance $\sigma_0^2 I$,

$$D_{\mathrm{KL}}(p_\phi \parallel q) = H(Z^{-\prime}) + \frac{\|\mu_\phi - \mu\|^2}{2\sigma_0^2} + \frac{d}{2}\log(2\pi\sigma_0^2), \tag{16}$$

where $\mu_\phi = \mathbb{E}[Z^{-\prime}]$ and $d$ is the feature dimension.

**Overall objective.**  Our final objective function holistically integrates all components of the framework established in Proposition 3.3. We minimise the combined loss:

$$\boxed{\mathcal{L}_{\mathrm{final}} = \mathcal{L}_{\mathrm{InfoNCE}} + \beta\,D_{\mathrm{KL}}(p_\phi \parallel q) - \lambda\,H(Z^{-\prime}) + \eta\,\mathcal{L}_{\mathrm{reg\_encoder}} + \gamma\,R(g_\phi)} \tag{17}$$

with $\lambda, \beta, \eta, \gamma > 0$. Here, the InfoNCE loss drives the primary representation learning task, while the KL-divergence term ensures that the transformations induced by SAIB maintain semantic consistency. The negative entropy term, $-\lambda\,H(Z^{-\prime})$, directly optimizes for greater diversity in the representation

---

[1]The query branch corresponds to the lower path in Fig. 1, whose encoder parameters are $f_{\theta_2}$.

space, serving as the practical objective for maximizing *incremental* entropy. Crucially, the novel encoder regularizer, $\eta \mathcal{L}_{\text{reg\_encoder}}$, operationalizes the entropy preservation principle central to our framework, ensuring that the diversity generated by SAIB is not lost during encoding. The final term, $\gamma R(g_\phi)$, is an optional weight-decay penalty on the SAIB module's parameters. This unified objective enables an end-to-end optimization of both entropy generation and preservation, yielding more robust representations.

# 4 EXPERIMENT & RESULT

## 4.1 EXPERIMENTAL SETUP

**Implementation Details** We conducted upstream self-supervised learning experiments on CIFAR-10 Krizhevsky & Hinton (2009), CIFAR-100 Krizhevsky & Hinton (2009), STL-10 Coates et al. (2011), and ImageNet Deng et al. (2009), followed by downstream evaluation on the PASCAL VOC. To ensure fair comparison, we used ResNet-based encoders across all experiments and fixed the random seed to 42 for reproducibility. For large-scale pretraining, we employed ResNet-50 as the backbone and followed the standard MoCo configuration on ImageNet for a consistent evaluation protocol. For ablation and scalability analysis, we used ResNet-18 with a batch size of 256 across CIFAR-10, CIFAR-100, STL-10, and ImageNet, enabling controlled comparisons under limited capacity settings. This phase primarily showcases the effectiveness of the proposed method under smaller batch settings across varying dataset distributions.

Table 1: The comparison of the proposed method with ResNet-50 as the backbone under different numbers of pre-training iterations. Using **BOLD** and Underline formatting to highlight the best and second results.

| Method | 100 ep | 200 ep | 400 ep | 800 ep | Batch Size |
|---|---|---|---|---|---|
| SimCLR (ICML'20) Chen et al. (2020a) | 66.5 | 68.3 | 69.8 | 71.1 | 4096 |
| SwAV (NeurIPS'20) Caron et al. (2020) | 66.5 | 69.1 | 70.7 | 71.0 | 4096 |
| MoCo-v2 (CVPR'20) Chen et al. (2020c) | 67.4 | 69.9 | 70.9 | 71.3 | 256 |
| SimSiam (ICCV'21) Chen & He (2021) | 68.1 | 70.0 | 70.8 | 71.7 | 256 |
| NNCLR (ICCV'21) Dwibedi et al. (2021) | 65.4 | 66.1 | 66.8 | 68.7 | 1024 |
| All4One (ICCV'23) Estepa et al. (2023) | 65.4 | 66.0 | 66.6 | 68.9 | 1024 |
| Matrix-SSL (ICML'24) Zhang et al. (2024) | **69.2** | 69.9 | 71.1 | 71.9 | 512 |
| Ours | 68.3 | **70.9** | **71.7** | **73.2** | **256** |

**Model Architectures** IE-CL was implemented on top of the MoCo framework, incorporating a momentum encoder and a symmetric contrastive loss as in SimCLR Chen & He (2021). A ResNet backbone with the classification head removed was used symmetrically on both anchor and query branches. The output features are 256-dimensional, obtained by global average pooling. Each branch uses a symmetric three-layer projector with an MLP-BN architecture. The hidden dimension is set to 4096, and the final projection is 512-dimensional. The anchor encoder and SAIB module are updated via backpropagation, while the query encoder is updated using momentum-based moving averages. The pseudo-code of IE-CL is shown in Appendix-Algorithm 1.

**Optimization and Hyperparameters** We trained IE-CL using AdamW with a batch size of 256, a base learning rate of 0.3, weight decay of 1e-5, and momentum of 0.9. Learning rates were scheduled via cosine annealing. The momentum coefficient $m$ for the momentum encoder was set to 0.9. The regularization weights for our final objective (Eq. 17) were configured as $\lambda = 0.2$ for entropy maximization, $\beta = 0.09$ for the KL-divergence, and $\gamma = 1e - 4$ for the SAIB weight decay. Crucially, the entropy-preserving encoder regularizer, $\mathcal{L}_{\text{reg\_encoder}}$, was implemented by applying Spectral Normalization to every convolutional layer of the encoder $f_\theta$, and its corresponding weight was set to $\eta = 1.0$ as it is an architectural constraint rather than a loss term. For linear evaluation, we used SGD with batch sizes of 512, learning rate of 0.03, momentum of 0.9, and weight decay of 1e-5. Cosine annealing was also used for scheduling. The linear classifier was trained for 200 epochs, and we report the final epoch accuracy. All experiments were conducted on 8 × NVIDIA Tesla V100 GPUs (32GB), using PyTorch 1.13 and Python 3.8.

Table 2: Comparison of self-supervised learning methods on various datasets (left) and segmentation/detection performance on PASCAL VOC2012 (right).

(a) Comparison based on **ResNet-18** with batch size is 256.

| Method | CIFAR-10 | CIFAR-100 | STL-10 | ImageNet |
|---|---|---|---|---|
| DeepCluster (ECCV'18)Caron et al. (2018) | 84.3 | 50.1 | 79.1 | 41.1 |
| SimCLR (ICML'20)Chen et al. (2020a) | 91.1 | 65.3 | 90.1 | 52.4 |
| MoCo-v2 (CVPR'20)He et al. (2020) | 91.3 | 68.3 | 88.9 | 52.5 |
| BYOL (NeurIPS'20)Grill et al. (2020) | 91.9 | 69.2 | 91.3 | 53.1 |
| SimSiam (ICCV'21)Chen & He (2021) | 91.2 | 64.4 | 90.5 | 33.2 |
| W-MSE (ICML'21)Ermolov et al. (2021) | 90.6 | 64.5 | 87.7 | 47.2 |
| MoCo-v3 (ICCV'21)Chen et al. (2021) | 91.8 | 68.8 | 91.4 | 56.1 |
| S3OC (TNNLS'22)Li et al. (2024) | 91.0 | 65.2 | 91.4 | - |
| MinEnt (PR'23)Li et al. (2023) | 90.8 | 66.1 | 91.5 | - |
| Light-MoCo (ICME'23)Lin et al. (2023) | - | - | - | 57.9 |
| Ours | **92.1** | **69.5** | **91.9** | **59.4** |

(b) Results on PASCAL VOC2012 with ResNet-50 SSL pretrained.

| Pretrained | mIoU | mAP |
|---|---|---|
| Supervised | 76.91 | 73.76 |
| Random | 38.35 | 40.90 |
| SimCLR | 76.74 | 73.17 |
| MoCo-v2 | 77.32 | 73.92 |
| SimSiam | 77.09 | 73.45 |
| Ours | **78.12** (↑ 1.21) | **74.41** (↑ 0.65) |

## 4.2 MAIN RESULTS

**Linear Evaluation** We adopt the standard linear evaluation protocol Chen et al. (2020a); Grill et al. (2020); He et al. (2022), in which the pretrained anchor encoder is frozen, and a linear classifier is trained on top. The anchor encoder, with the backbone network parameters frozen, is used for the linear evaluation process. A linear layer is appended and trained using supervised signals while keeping the backbone fixed. Training data is augmented via random horizontal flipping, random cropping to $224 \times 224$, and layer normalization. For evaluation, input images are resized from $256 \times 256$ to $224 \times 224$. Table 1 reports Top-1 accuracy after IE-CL pretraining on ImageNet using ResNet-50 over 100, 200, 400, and 800 epochs. Table 2 shows linear probe results on other datasets using ResNet-18 (trained for 300 epochs on ImageNet and 1,000 on smaller datasets). IE-CL consistently outperforms previous baselines across all settings.

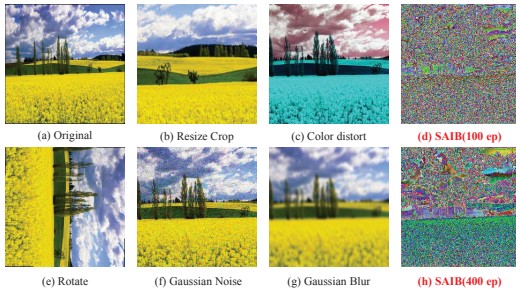

Figure 2: Illustration of the data augmentation operators studied. The non-isometric transformation operator SAIB has learnable parameters, enabling non-prior augmentation for contrastive learning. Visualizing changes from 100 epochs (d) to 400 epochs (h) shows that KL divergence effectively constrains incremental entropy, preventing collapse.

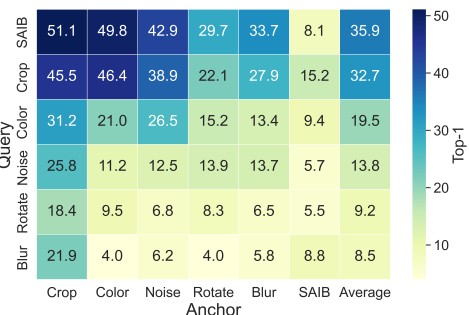

Figure 3: Ablation tests the relationship between SAIB and the previous pretext task. The image was resized to 224×224, and augmentation strength settings from Chen et al. (2020a) were applied, followed by two-by-two tests with SAIB placed on both sides of the contrastive learning.

**Transfer Learning** To assess the transferability of the learned representations, we evaluate IE-CL on two downstream tasks from PASCAL VOC 2012 Everingham et al. (2010): object detection and semantic segmentation. We use Faster R-CNN Ren et al. (2016) for object detection and DeepLab-v3 Chen et al. (2018) for segmentation, both with ResNet-50 backbones pretrained via IE-CL. For segmentation, training samples are augmented with random cropping and contrast-based enhancement. Adam is used with a learning rate of $3 \times 10^{-4}$. For detection, the model is trained using SGD with a learning rate of $1 \times 10^{-4}$.

**Augmentation Dependency** As SAIB is implemented at the data loading stage (see Appendix Algorithm 1), it can be interpreted as a learnable augmentation layer, contrasting with traditional pretext-based augmentation schemes used in prior contrastive methods. Figure 3 presents comparative

results for various augmentation strategies on ImageNet-100, under 100 epochs of pretraining and linear evaluation. Our method exhibits robust performance gains under limited augmentation.

Table 3: Ablation study of the IE-CL components on ImageNet-1k using MoCo-v2 with a ResNet-18 backbone. We incrementally add our proposed components: the SAIB module for entropy generation, KL regularization for semantic consistency, and an Encoder Regularizer (implemented via Spectral Normalization) for entropy preservation.

| Configuration | SAIB | KL Reg. | Encoder Reg. | Top-1 |
|---|---|---|---|---|
| MoCo-v2 (Baseline) | ✗ | ✗ | ✗ | 52.50 |
| + Entropy Generation | ✓ | ✗ | ✗ | 58.80 |
| + Semantic Consistency | ✓ | ✓ | ✗ | 59.15 |
| **IE-CL (Full Framework)** | ✓ | ✓ | ✓ | **59.41** |

**Ablation Study** To assess the contribution of each component in IE-CL, we performed an ablation study based on a MoCo-v2 baseline with a ResNet-18 backbone on ImageNet-1k. As shown in Table 3, progressively adding the core modules in Proposition 3.3 yields consistent gains. Introducing the SAIB module for *entropy generation* produces the largest improvement, confirming the benefit of maximizing input-level incremental entropy, while the KL-divergence term for *semantic consistency* further enhances performance by mitigating distributional drift. Crucially, the final row demonstrates that incorporating our proposed *entropy preservation* mechanism via an encoder regularizer ('Encoder Reg.') provides an additional performance boost on top of the already strong SAIB+KL configuration. This result provides strong empirical evidence for the central tenet of our framework: that optimal performance is achieved by jointly optimizing for both entropy generation at the input and entropy preservation through the encoder. We also found that cascading multiple SAIB modules offered diminishing returns, shown in Table 4, thus we use a single module in our main configuration.

**Plug and Play** Table 5 demonstrates the plug-and-play ability of SAIB when integrated into other self-supervised learning frameworks on ImageNet-100, including non-contrastive methods such as BYOL and SimSiam. At this point, SAIB is placed on the *Target* side, similar to its placement on the *Anchor* side in contrastive learning. We further visualize entropy gains during training in Figure 4 and 5, showing accelerated convergence and performance improvement attributed to SAIB.

Table 4: Ablation on the number of cascaded SAIB modules within the full IE-CL framework. Performance slightly degrades with more than one module, indicating diminishing returns.

| Configuration | SAIB Cascade | Top-1 |
|---|---|---|
| IE-CL (Full Framework) | 1x | **59.41** |
| IE-CL with more modules | 2x | 58.62 |
|  | 3x | 58.71 |

Table 5: Based on the theory of maximizing incremental information entropy with non-isometric transformations, SAIB can be seamlessly integrated to enhance other self-supervised paradigms.

| Method | Top1 | Batch Size | Epoch |
|---|---|---|---|
| MoCo-v2 | 66.29 | 256 | 200 |
| BYOL | 67.95 | 256 | 200 |
| SimCLR | 63.34 | 256 | 200 |
| SimSiam | 66.25 | 256 | 200 |
| MoCo-v2 + SAIB | 67.54 (↑ 1.25) | 256 | 200 |
| BYOL + SAIB | 68.76 (↑ 0.81) | 256 | 200 |
| SimCLR + SAIB | 64.02 (↑ 0.68) | 256 | 200 |
| SimSiam + SAIB | 66.97 (↑ 0.72) | 256 | 200 |

## 5 DISCUSSION, LIMITATION, CONCLUSION AND FUTURE WORK

This work introduces *Sample Incremental Information Entropy* and presents a new framework, IE-CL, to advance mutual information maximization in contrastive learning. It addresses the critical challenge of the encoder information bottleneck by jointly optimizing for *entropy generation*, via a novel learnable transformation module (SAIB), and *entropy preservation*, via an explicit encoder regularizer. Our approach yields consistent improvements across various datasets, though several aspects merit further study. SAIB operates at the patch level and induces local pixel-space variations, which preserve semantic consistency but may limit expressiveness in modeling complex structures or higher-resolution tasks. Its reliance on convolutional priors also raises challenges for extension to vision transformers or Vision Kolmogorov–Arnold NetworksYang et al. (2026). Accordingly, exploring

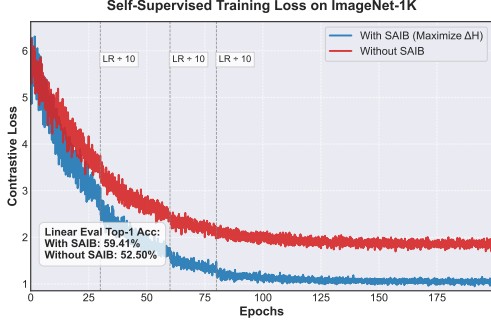
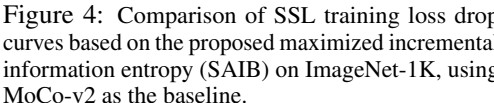
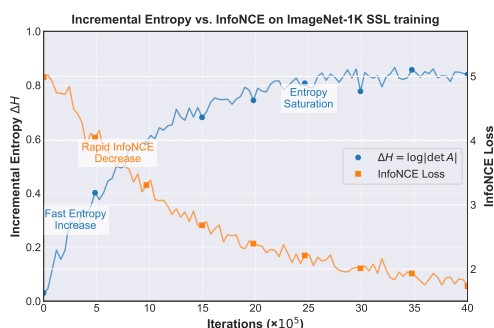

Figure 4: Comparison of SSL training loss drop curves based on the proposed maximized incremental information entropy (SAIB) on ImageNet-1K, using MoCo-v2 as the baseline.

Figure 5: The variation of the incremental entropy $\Delta H(X)$ on the Query side and InfoNCE throughout the iterations is shown.

broader cross-domain integration strategies and adapting them to diverse visual backbonesYang et al. (2025) may offer a promising direction for future research. Nonetheless, the core principle of IE-CL, explicitly modeling and maximizing sample entropy, provides a principled perspective for augmentation design and entropy-aware optimization, enriching representation diversity and deepening the information-theoretic understanding of self-supervised learning.

## ACKNOWLEDGMENTS

This work was supported by the National Natural Science Foundation of China under Grant 62576216, Guangdong Provincial Key Laboratory under Grant 2023B1212060076, and also supported by the Intelligent Computing Center of Shenzhen University.

## ETHICS STATEMENT

This work does not involve human subjects, personally identifiable information, or sensitive medical data. All experiments are conducted on publicly available benchmark datasets (CIFAR-10/100, STL-10, ImageNet, and PASCAL VOC), which are widely adopted in the research community. We adhere strictly to the ICLR Code of Ethics and the licensing terms of the datasets used. Our proposed method, IE-CL, is intended for advancing self-supervised learning research in computer vision and does not present foreseeable risks of harmful misuse. We disclose all relevant implementation details, maintain academic integrity, and ensure that our research complies with ethical standards of reproducibility, transparency, and fairness.

## REPRODUCIBILITY STATEMENT

We have made extensive efforts to ensure the reproducibility of our results. A detailed description of the proposed method, IE-CL, including the theoretical derivations (Section 3), algorithm design (SAIB module), and the overall objective function, is provided in the main text. The experimental setup, datasets, and evaluation protocols are described in Section 4, with optimization hyperparameters and implementation details explicitly listed. Additional pseudo-code and derivations are included in the Appendix. We will release the source code, training scripts, and configuration files after the paper is accepted, as supplementary materials to enable full reproducibility. Random seeds and hardware specifications are also reported to facilitate consistent replication of our experiments.

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

SUPPLEMENTARY MATERIALS

## A  THE USE OF LARGE LANGUAGE MODELS (LLMS)

In preparing this manuscript, we employed large language models (LLMs) solely for language polishing and grammar refinement. The LLMs were not involved in idea generation, theoretical development, algorithm design, experimental implementation, or result analysis. All technical content, experiments, and conclusions presented in this work are entirely the contribution of the authors.

## B  THEORETICAL JUSTIFICATION FOR THE IE-CL FRAMEWORK

This appendix provides a detailed theoretical argument for our proposed Incremental Entropy Contrastive Learning (IE-CL) framework. We first establish why maximizing the entropy of negative sample representations, $H(Z'^-)$, is a desirable objective within the InfoNCE framework. We then use the Data-Processing Inequality to formally demonstrate why naively maximizing input-level entropy is insufficient due to the information bottleneck of deep encoders. Finally, we show how our full IE-CL objective function provides a principled and complete solution to this challenge.

### B.1  THE GOAL: MAXIMIZING NEGATIVE ENTROPY FOR BETTER CONTRASTIVE LEARNING

The standard InfoNCE loss for a positive pair $(z, z^+)$ and a set of $N-1$ negative samples $\{z_k^-\}_{k=1}^{N-1}$ drawn from a distribution $q(z^-)$ is:

$$\mathcal{L}_{\text{InfoNCE}} = -\mathbb{E}\left[\log \frac{\exp(s(z, z^+)/\tau)}{\exp(s(z, z^+)/\tau) + (N-1)\mathbb{E}_{z^- \sim q}[\exp(s(z, z^-)/\tau)]}\right] \quad (18)$$

Our core premise is that increasing the entropy of the negative distribution, $H(Z'^-)$, where $q$ is the distribution of $Z'^-$, makes the contrastive task more challenging and thus compels the model to learn better representations. Let's formalize this.

The denominator of the InfoNCE loss can be seen as a partition function. A higher entropy $H(Z'^-)$ implies that the negative samples $z^-$ are more diverse and spread out in the representation space. This increased diversity makes it statistically more likely for some negative samples to be close to the anchor $z$, thus increasing the expected value of the negative scores, $\mathbb{E}_{z^- \sim q}[\exp(s(z, z^-)/\tau)]$.

This directly increases the value of the denominator, which in turn increases the InfoNCE loss. To compensate for this more difficult learning signal (i.e., to minimize the loss), the optimizer is forced to adapt the encoder parameters $(\theta_1, \theta_2)$ to create a sharper separation. This is primarily achieved by increasing the similarity of the positive pair, $s(z, z^+) \uparrow$.

An increased positive pair similarity implies that given an anchor $z$, its positive counterpart $z^+$ becomes more predictable. In information-theoretic terms, this corresponds to a reduction in the conditional entropy, $H(Z^+|Z) \downarrow$. According to the definition of mutual information, $I(Z; Z^+) = H(Z^+) - H(Z^+|Z)$, a decrease in conditional entropy (while the marginal entropy $H(Z^+)$ is kept non-trivial to prevent collapse) leads to an increase in the mutual information, $I(Z; Z^+) \uparrow$. This is the ultimate goal of InfoNCE-based contrastive learning.

Thus, we have established the following desirable causal relationship:

$$\max H(Z'^-) \implies \min H(Z^+|Z) \implies \max I(Z; Z^+) \iff \min \mathcal{L}_{\text{InfoNCE}} \quad (19)$$

This confirms that maximizing the entropy of negative representations is a valid and principled objective for improving contrastive representation learning.

### B.2  THE CHALLENGE: THE INFORMATION BOTTLENECK IN DEEP ENCODERS

Having established our goal, the naive strategy would be to simply maximize the entropy at the input of the encoder, $H(X'^-)$, using our SAIB module, $g_\phi$. However, this approach is fundamentally flawed because it ignores the transformative effect of the deep encoder, $f_\theta$.

The Data-Processing Inequality for differential entropy provides a formal tool to analyze this. Let $X'^- = g_\phi(X^-)$ be the transformed input. The entropy of the final representation, $Z'^- = f_\theta(X'^-)$, is bounded by the entropy of its input $H(X'^-)$:

$$H(Z'^-) = H(f_\theta(X'^-)) \leq H(X'^-) + \mathbb{E}_{p(x')}[\log|\det J_{f_\theta}(x')|] \tag{20}$$

where $J_{f_\theta}(x')$ is the Jacobian of the encoder function $f_\theta$ evaluated at $x'$.

This inequality reveals the core challenge. While our SAIB module is designed to maximize $H(X'^-)$, the second term, $\mathbb{E}[\log|\det J_{f_\theta}|]$, which depends entirely on the encoder, can be a large negative value. This occurs if the encoder acts as a severe **information bottleneck**, aggressively compressing or collapsing its input space. In such a scenario, the entropy gained at the input level via SAIB would be nullified by the entropy lost during the encoding process.

Therefore, we conclude that maximizing the input-level incremental entropy $\Delta H(X^-)$ (and thus $H(X'^-)$) is a **necessary but not sufficient** condition. To robustly increase the final representation entropy $H(Z'^-)$, a mechanism to control the encoder's information-compressing behavior is essential.

### B.3 THE IE-CL SOLUTION: A SYNERGISTIC OPTIMIZATION FRAMEWORK

Our IE-CL framework provides a complete solution by reformulating the objective to jointly optimize both entropy generation and preservation. We re-state our final loss function from the main text:

$$\mathcal{L}_{\text{final}} = \mathcal{L}_{\text{InfoNCE}} + \beta D_{\text{KL}}(p_\phi||q) - \lambda H(Z'^-) + \eta \mathcal{L}_{\text{reg\_encoder}} \tag{21}$$

Let's analyze how this objective creates an optimization landscape that solves the challenge described in Sec. B.2. The goal of the optimizer is to minimize $\mathcal{L}_{\text{final}}$, which is dominated by the term $-\lambda H(Z'^-)$, effectively becoming an objective to maximize $H(Z'^-)$. To achieve this, the optimizer can adjust the parameters of SAIB ($\phi$) and the encoder ($\theta$).

1. **Optimizing SAIB ($\phi$)**: To maximize the final entropy $H(Z'^-)$, the optimizer is incentivized to maximize the input entropy $H(X'^-)$, as established by the bound in Eq. 20. The SAIB module, $g_\phi$, is specifically designed for this task. As shown in the appendix, its design as a volume-expanding map ($|\det J_{g_\phi}| > 1$) directly translates to maximizing the incremental entropy $\Delta H(X^-)$. This is the **entropy generation** part of our framework.

2. **Optimizing the Encoder ($\theta$)**: The term $\eta \mathcal{L}_{\text{reg\_encoder}}$ directly constrains the encoder. By implementing this regularizer via **Spectral Normalization**, we constrain the Lipschitz constant of the encoder's layers. A smaller Lipschitz constant leads to a "smoother" transformation, which in turn prevents the Jacobian determinant term $\mathbb{E}[\log|\det J_{f_\theta}|]$ from becoming excessively negative. This term directly counteracts the information bottleneck, serving as the **entropy preservation** part of our framework.

3. **Semantic Constraint ($D_{KL}$)**: The KL-divergence term acts as a crucial regularizer on SAIB, ensuring that the entropy maximization process does not push the transformed samples $X'^-$ into a semantically meaningless or out-of-distribution space.

In conclusion, the IE-CL objective function does not assume a naive carry-over of entropy. Instead, it creates a synergistic system where the only effective way for the optimizer to maximize the final representation entropy $H(Z'^-)$ is to **simultaneously** use SAIB to generate rich input entropy and constrain the encoder to faithfully preserve it. This provides a principled and robust solution to learning diverse representations for contrastive learning.

## C JACOBIAN DETERMINANT OF THE SAIB

### C.1 DETAILS OF THE SAIB

**Block definition.** Let $x \in \mathbb{R}^D$ be the flattened patchified tensor. Within a fixed ReLU activation pattern the block acts linearly:

$$f(x) = x + A\,x, \qquad A := W_4 M_3 W_3 M_2 W_2 M_1 W_1, \tag{22}$$

where

- $W_1 \in \mathbb{R}^{D \times D}$, $W_3 \in \mathbb{R}^{2D \times 2D}$, $W_4 \in \mathbb{R}^{D \times 2D}$ are $1 \times 1$-convs;

- $W_2 \in \mathbb{R}^{2D \times D}$ is a $3 \times 3$-conv that *doubles* the channel dimension;

- $M_i$ are diagonal $0/1$ masks coming from ReLU derivatives.

**Step 1: A lower bound on $\|A\|_2$.**  Because $W_2$ maps $\mathbb{R}^D \to \mathbb{R}^{2D}$ with i.i.d. Gaussian initialisation of variance $2/\mathrm{fan}_{\mathrm{in}}$, random matrix theory gives

$$\Pr\big[\sigma_{\max}(W_2) \geq \sqrt{2}\big] = 1. \tag{23}$$

All other $W_i$ are square and full rank by construction, so $\|A\|_2 \geq \sqrt{2} \; \|W_4 M_3 W_3 M_1 W_1\|_2 > 1$ almost surely.

**Step 2: Singular values of the Jacobian.**  The Jacobian of $f$ is

$$J = I + A. \tag{24}$$

Let $u$ be the right singular vector of $A$ associated with $\sigma_{\max}(A) =: s > 1$. Then

$$\|Ju\|_2 = \|u + Au\|_2 \geq \big\|Au\big\|_2 - \|u\|_2 = s - 1 > 0, \tag{25}$$

and by triangle inequality also $\|Ju\|_2 \geq 1 + s$. Hence the largest singular value of $J$ satisfies $\sigma_{\max}(J) \geq 1 + s > 2$.

**Step 3: Determinant strictly greater than $1$.**  Since $J$ is the sum of identity and a matrix of full column rank, every singular value of $J$ is $\geq 1$ (see Weyl's monotonicity theorem). With at least one singular value $> 2$ we get

$$|\det J| = \prod_{k=1}^{D} \sigma_k(J) > 2 \times 1^{D-1} > 1. \tag{26}$$

Therefore the block is *locally volume-expanding* almost everywhere, and its differential entropy change $\Delta H = \mathbb{E}[\log |\det J|] > 0$.

**Remark.**  Even if some ReLU masks set entire channels to zero, the $2\times$ expansion ensures that at least one singular value of $A$ remains $> 1$ with high probability, keeping the argument intact.

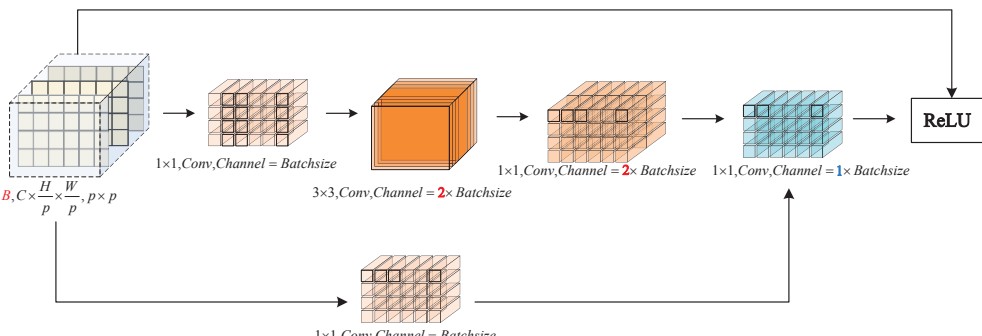

Figure 6: Structure of Sample Augmentation Incremental Block (**SAIB**). Note that due to the patching of the 3-channel image, the batch occupies the position of the original channel. Therefore, it is possible to drive the inter-batch information to communicate by changing the original convolutional channels. All the convolutions in this block are cascaded with BN layers and SwishRamachandran et al. (2017) to achieve nonlinear augmentation capability. And then convolved and non-linearly processed, and finally reconstructed back to the original position through positional coding.

# D TRAINING COST

We show a comparison of the training time consumed for the proposed strategies in Figure 7 and Figure 8, respectively. Figure 7 shows the different methods at 256 batch setting a with resnet50 as backbone on ImageNet-1k The time required to train one epoch. All parameters were kept at the optimal settings declared at the time of their release, and time spent was evaluated using mixed precision on $8 \times$ V100 (32G).

Figure 8 illustrates the additional computational time consumption associated with the SAIB plug-and-play existing approach. Due to the differences in the self-supervised paradigms, we observe that for the encoder half-update paradigm (MoCo-v2, SimSiam, BYOL), adding SAIB to maximise the incremental information entropy results in only a slight additional computation time (within 10%), whereas for the full-parameter update approach that relies heavily on the batch scaling to function (SimCLR), adding SAIB increases the training time by 12.3%. Overall, SAIB is able to balance the performance improvement of the model with the increased training time.

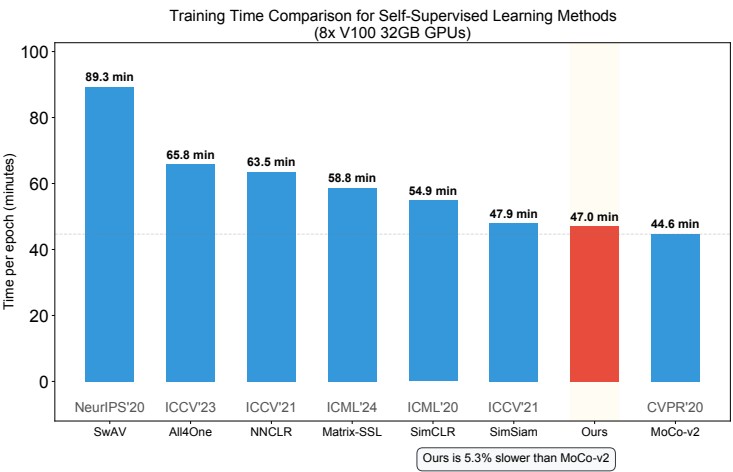

Figure 7: Comparison of the time taken by different methods to train an epoch on ImageNet-1k with batch of 256. The proposed IE-CL, although it includes an additional non-isometric transform module SAIB, still spends less training cost compared to the previous methods because it uses momentum to update the Query encoder.

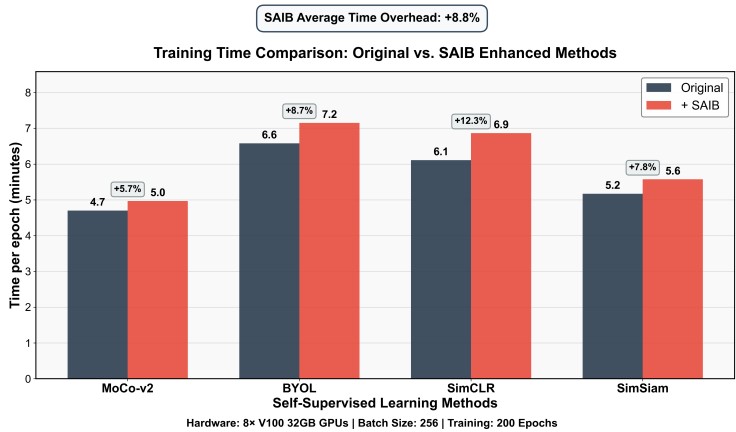

Figure 8: As a plug-and-play module, SAIB enhances the performance of existing contrastive learning methods with limited additional computational overhead. Overall, it achieves effective performance gains within an acceptable increase in training time—on average, approximately 8.8% more—compared to the original models (see Table 5 in the main text).

# E  PSEUDO CODE

**Algorithm 1:** PyTorch pseudo-code of IE-CL (Corrected)

```python
# Q: anchor encoder (updated by backprop)
# K: query encoder (updated by momentum)
# m:  momentum hyperparameter for K
# ctr:  contrastive loss function (e.g., InfoNCE)
# SAIB: sample augmentation incremental block
# optimizer:  updates Q and SAIB parameters
# H: entropy estimator

# Initialize K's parameters from Q's
K.load_state_dict(Q.state_dict())

for x in loader:
  # Create two augmented views
  x_anchor, x_query = aug(x), aug(x)

  # Apply SAIB to the query view to increase entropy
  x_query_transformed = SAIB(x_query)

  # -- Forward Pass --
  # Q computes features for anchor and transformed query
  q_anchor = Q(x_anchor)
  q_query_transformed = Q(x_query_transformed)

  # K computes features for transformed query (no gradients)
  with torch.no_grad():
     k_query = K(x_query_transformed)

  # -- Loss Calculation (matches Equation 25) --
  # 1.  InfoNCE Loss
  𝓛_InfoNCE = ctr(q_anchor, k_query)
  # 2.  KL divergence for regularization
  𝓛_KL = KL_Loss(q_query_transformed.detach(), q_anchor)
  # 3.  Incremental Entropy Maximization
  𝓛_entropy_max = -H(q_query_transformed)

  # Total loss
  loss = 𝓛_InfoNCE + β * 𝓛_KL + λ * 𝓛_entropy_max

  # -- Backward Pass & Optimizer Step --
  loss.backward()
  optimizer.step()
  optimizer.zero_grad()

  # -- Momentum Update K --
  with torch.no_grad():
     for param_q, param_k in zip(Q.parameters(), K.parameters()):
        param_k.data = param_k.data * m + param_q.data * (1.0 - m)
```

