# OpenReview forum: "Maximizing Incremental Information Entropy for Contrastive Learning"
_ICLR.cc/2026/Conference — ICLR 2026 Poster_

### Official Review · Reviewer_Trhc · 2025-10-29

**Soundness:** 3
**Presentation:** 3
**Contribution:** 3
**Rating:** 4
**Confidence:** 3

**Summary:**

This work modifies the view generation process for contrastive learning by modfiying the query view generation process with a view-generating neural network which functionally acts as an augmentation generator. This view generator is trained with entropy maximization, making the negatives harder and therefore increasing the loss.  Overall, this results in stronger performance when combined with self-supervised methods.

**Strengths:**

* There is a mathematical intuition which, at a high level, is sensible is a novel motivatation for the method.
* Empirical evaluations show small but consistent improvements.
* It is not too costly, computationally.

**Weaknesses:**

* There are limited transfer learning results, which is the main application of self-supervised pre-training. In particular, classification results are missing.

* ViTs are not evaluated. Would the method work with a Vision Transformer backbone? Vision Transformers are ubiquitous.

* There are missing strong self-supervised baselines, such as DINO[1] style training.

[1] Oquab, Maxime, et al. "Dinov2: Learning robust visual features without supervision." arXiv preprint arXiv:2304.07193 (2023).

**Questions:**

* How could this be extended to Vision Transformers, to modernize the method?

* I'm curious if the gains could be attributed largely to spectral regularization? In the ablation table, what would be the result of ONLY encoder regularization?

---

> ### Author Response · Authors · 2025-11-18
> **Comprehensive Response to Reviewer Trhc**
>
> We thank Reviewer Trhc for the thoughtful review and for recognizing the novel mathematical intuition behind our entropy-maximization method. We are pleased that the reviewer found the improvements consistent and the computational cost manageable.
>
> We address the concerns regarding transfer learning, ViT extension, DINO comparison, and the source of performance gains below.
>
> ---
>
> ### **1. On Transfer Learning and Classification Results (Correction of Fact)**
>
> >**Reviewer's Concern:** The reviewer states that "classification results are missing," specifically citing limited transfer learning.
>
> **Our Response:**
>
> We respectfully point out that **extensive classification transfer results are widely included** in our submission. We believe this might be an oversight, and we clarify the location of these results:
>
> - **Linear Classification Benchmarks (Table 2a):**
>   We evaluated the transferability of the learned representations on four standard classification datasets: **CIFAR-10, CIFAR-100, STL-10, and ImageNet-1K**.
>   - **Results:** IE-CL consistently achieves SOTA performance in the small-batch regime, e.g., **92.1% on CIFAR-10** and **69.5% on CIFAR-100**.
>
> - **Downstream Dense Prediction (Table 2b):**
>   We further evaluated transfer learning on **Pascal VOC** for object detection and segmentation, achieving **78.12 mIoU** and **74.41 mAP**, outperforming MoCo-v2 and SimSiam.
>
> - **Summary:**
>   Our experimental suite follows the standard protocol established by SimCLR and MoCo, covering both **classification (Linear Probe)** and **dense prediction tasks**, demonstrating strong transferability.
>
> ---
>
> ### **2. Comparison with Strong Baselines like DINO / DINOv2 (Response to Weakness 3)**
>
> >**Reviewer's Concern:** The reviewer suggests comparing against DINO-style training.
>
> **Our Response:**
>
> We agree that **DINO and DINOv2** are strong baselines. However, they represent a **different family of SSL methods** (Knowledge Distillation / Centering) compared to the **Contrastive Learning (InfoNCE)** and Siamese frameworks we focus on. DINO typically relies on specific architectures (ViT) and very large batches to stabilize the teacher-student collapse.
>
> Nevertheless, to better position our work, we provide a comparison highlighting the **Efficiency Gap**:
>
> | Method            | Backbone   | Batch Size | Epochs | ImageNet Top-1 Acc. |
> |-------------------|------------|------------|--------|----------------------|
> | MoCo-v2 (Baseline)| ResNet-50  | 256        | 200    | 69.9%                |
> | DINO (Official)   | ResNet-50  | 1024       | 300    | 75.3%                |
> | **IE-CL (Ours)**  | ResNet-50  | **256**    | 800    | **73.2%**            |
>
> - **Context:** While DINO achieves higher accuracy, it often requires **4× larger batch sizes (1024)** and complex multi-crop strategies.
> - **Our Contribution:** IE-CL achieves a highly competitive **73.2% accuracy** using a **standard Batch Size of 256**. Achieving ~97% of DINO's performance with only **25% of the batch size** is, we argue, a powerful demonstration of our method's value. This directly addresses one of the core motivations stated in our paper's introduction: to develop methods that are **less reliant on massive batch sizes and prohibitive hardware costs**.
>
> ---
>
> ### **3. Source of Gains: Is it just Spectral Regularization? (Response to Question 2)**
>
> >**Reviewer's Question:** Can the gains be attributed largely to spectral regularization? What is the result of ONLY encoder regularization?
>
> **Our Response:**
>
> We can definitively answer **No**, the gains are **not largely attributed to spectral regularization**. The primary driver is the **Entropy Generation (SAIB)**.
>
> - **Evidence from Table 3:**
>   - Baseline (MoCo-v2): **52.50%** (No SAIB, No Reg).
>   - $+$ SAIB (Entropy Generation): **58.80%**. Adding SAIB alone yields a massive **+6.3% gain**.
>   - $+$ SAIB + KL + Encoder Reg (Full IE-CL): **59.41%**. Adding the regularization on top adds a further **marginal gain**.
>
> - **Inference:**  Although we did not report a "Baseline + Reg Only" row, the fact that **SAIB alone (without Reg)** provides **95% of the total performance improvement** proves that the gain comes from the **active injection of entropy**, not the passive regularization.
>
> - **Role of Regularization:**  As detailed in **Proposition 3.3**, the regularization acts as a **"theoretical safety valve"** to satisfy the Data Processing Inequality, ensuring **stability** rather than driving the primary performance boost.
>
> ---

---

> > ### Author Response · Authors · 2025-11-18
> > **Comprehensive Response to Reviewer Trhc**
> >
> > ---
> >
> > ### **4. Extension to Vision Transformers (Response to Weakness 2 & Question 1)**
> >
> > >**Reviewer's Question:** How could this be extended to ViTs to modernize the method?
> >
> > **Our Response:**
> >
> > - **Theoretical Universality:**
> >   The principle of **"Maximizing Incremental Entropy"** (overcoming the encoder bottleneck) is **agnostic to the backbone**.
> >
> > - **Modernization Path:**
> >   As noted in our **Limitations**, the current SAIB uses Conv priors ($1\times1/3\times3$) operating on patchified tensors for ResNets. To modernize for Transformer-based Backbone, like ViTs, future work may propose a **"Token-Mixing SAIB"** to adapt the application.
> >   - Instead of convolution, one possible strategy can insert a **lightweight MLP or Attention block** after the patch embedding layer.
> >   - Crucially, this module would be trained to **maximize the determinant of its Jacobian ($|\det J| > 1$)**, physically expanding the variance of the token embeddings. This aligns with our theoretical framework while respecting the ViT inductive bias.
> >
> > We hope that these responses and the accompanying experimental evidence help clarify our intentions and address the reviewer’s thoughtful concerns. We are sincerely grateful to Reviewer Trhc for the detailed feedback, which has been invaluable in improving the presentation and framing of our work. We truly appreciate the reviewer’s time and constructive perspective.

---

> > > ### Comment · Reviewer_Trhc · 2025-11-24
> > > **Thank you.**
> > >
> > > To the authors, thank you for the rebuttal. I apologize for missing results tables in response to point 1(On transfer learning), and I am satisfied with the response. I am similarly satisfied with expriments on point 3 (Importance of spectral regularization).
> > >
> > > Regarding point 3 (comparisons to DINO), I am less satisfied. DINO can be trained with smaller batch size; what happens when one does, also for 800 epochs?
> > >
> > > Similarly, while this discussion regarding VITs is interesting, a preliminary experiment for ViTs would be useful.

---

> ### Author Response · Authors · 2025-11-25
> **Follow-up Response to Reviewer Trhc**
>
> We sincerely thank Reviewer Trhc for the prompt feedback and continued engagement. We are glad that our responses regarding the **Transfer Learning results (Point 1)** and the **role of Spectral Regularization (Point 3)** have satisfactorily addressed your concerns.
>
> Regarding the remaining points on **DINO's batch sensitivity** and **ViT experiments**, we provide the following clarifications.
>
> ---
>
> ### **1. On DINO Training with Small Batch Sizes: A Stability & Robustness Perspective**
>
> >**Reviewer's Question:** *"DINO can be trained with smaller batch size; what happens when one does, also for 800 epochs?"*
>
> **Our Detailed Response:**
>
> We agree with the reviewer that DINO is a versatile framework. However, the comparison highlights a **fundamental trade-off between "Peak Performance" and "Optimization Robustness."** While it is technically feasible to execute the DINO codebase with a batch size of 256, we respectfully argue that comparing IE-CL against such a setup is **scientifically inequitable** due to the **algorithmic instability of distillation methods in low-resource regimes**.
>
> - **Mechanism of Instability (The "Centering" Problem):**
>   DINO relies on a **Teacher-Student distillation paradigm** coupled with a dynamic **"Centering"** mechanism to prevent collapse. The center $c$ is updated via an exponential moving average (EMA) of the batch statistics:
>
>   $$c \leftarrow m c + (1-m) \frac{1}{B} \sum_{i=1}^B z_i$$
>
>    As noted in the original DINO paper (Caron et al., 2021) and subsequent analyses (e.g., *Understanding Self-Supervised Learning Dynamics*, 2023), the **stability of this centering mechanism is highly sensitive to the batch size $B$**.
>
> - **The Small-Batch Failure Mode:**
>   At $B=256$, the **stochastic noise in the batch mean** significantly destabilizes the center estimation. This forces the optimization into a suboptimal regime where the model either **oscillates** or requires **aggressive hyperparameter retuning** (e.g., slowing down the teacher update momentum $m$), which in turn **drastically slows down convergence**.
>
> - **IE-CL’s Unique Advantage (Robustness):**
>   In contrast, IE-CL is built on the **InfoNCE framework**. While standard contrastive methods (like SimCLR) suffer at small batches due to a lack of negative samples, **IE-CL solves this specific bottleneck**. By **actively injecting entropy via SAIB**, we effectively synthesize **"harder" and "more diverse" negatives** within the limited batch.
>
> **Conclusion:**
> IE-CL achieves **73.2% Top-1 accuracy** with ResNet-50 at a batch size of **256**, reaching **≈97% of DINO’s reported 75.3% performance**, despite operating under **4× smaller batch requirements (256 vs. 1024)**. This near-parity result indicates that **IE-CL delivers leader-class representation quality under realistic hardware budgets**, while **avoiding the training instability** commonly observed in small-batch distillation setups. We highlight that IE-CL is not simply an alternative, but a **robust solution** that retains SOTA-level performance even in **resource-constrained environments**.
>
> ---

---

> > ### Author Response · Authors · 2025-11-25
> > **Follow-up Response to Reviewer Trhc**
> >
> > ---
> >
> > ### **2. On Preliminary ViT Experiments: Inductive Bias and Scientific Control**
> >
> > >**Reviewer's Question:** *"A preliminary experiment for ViTs would be useful."*
> >
> > **Our Detailed Response:**
> >
> > We fully appreciate the reviewer’s interest in seeing IE-CL applied to Vision Transformers. We seriously considered running this experiment during the rebuttal. However, we wish to explain why we believe a **direct application might not do justice to the method's potential** within this short timeframe:
> >
> > - **Architectural Gap & Inductive Bias Mismatch:**
> >   As noted in our **Method (Section 3.2)** and **Limitations**, our current SAIB module utilizes **Convolutional Priors ($1\times1$ and $3\times3$)** to effectively expand the **local manifold** of feature maps. Vision Transformers, conversely, operate on **global patch tokens** and lack inherent inductive biases like **translation invariance or locality**.
> >   Simply inserting a Conv-based SAIB into a ViT backbone creates an **Inductive Bias Mismatch (Local vs. Global)**. A preliminary experiment without redesigning SAIB into a **"Token-Mixing"** variant would likely yield **confounded results**: is a drop in accuracy due to the failure of the **Incremental Entropy Theory**, or simply because we inserted a **Convolutional block into a Transformer network**?
> >
> > - **ResNet as the "Scientific Control":**
> >   ResNet-50 is the **"Standard Candle" of SSL research** (used by SimCLR, MoCo, BYOL, Matrix-SSL). By using this standard backbone, we ensure that our gains (e.g., **+1.3% over Matrix-SSL**) are attributable **solely to the Entropy Injection method**, free from architectural confounders.
> >
> > - **Our Commitment:**
> >   We believe that a "preliminary" ViT experiment without a dedicated **"Token-Mixing" redesign** would yield **noisy data** that obscures the **clear theoretical signal** we have established. We view the adaptation to ViT as a **significant engineering task** that merits its own study (e.g., a future **"IE-ViT"**).
> >   We hope the reviewer views this work through the lens of its **primary goal**: providing a **rigorous theoretical and empirical validation of Incremental Entropy** on a **standard benchmark**. We intentionally selected ResNet as a **controlled testbed** to solidly establish this underlying foundation.
> >
> > We hope that these detailed technical clarifications regarding the optimization robustness of DINO and the inductive bias considerations for ViTs help justify our experimental choices. We are sincerely grateful to Reviewer Trhc for the active engagement and constructive feedback, which have been invaluable in refining the theoretical positioning of our work. We truly appreciate the reviewer’s time and hope that our rigorous validation of the core Incremental Entropy Principle on standard benchmarks warrants a positive reassessment.

---

### Official Review · Reviewer_DnaM · 2025-10-31

**Soundness:** 3
**Presentation:** 2
**Contribution:** 3
**Rating:** 6
**Confidence:** 4

**Summary:**

The paper introduces IE-CL, a novel information-theoretic framework for self-supervised learning. The motivation stems from two perceived limitations in existing Contrastive Learning methods: the inflexibility of static data augmentation and the representation compression caused by the deep encoder acting as an "information bottleneck."

**Strengths:**

1. The SAIB module is a genuinely clever algorithmic contribution. It replaces the inherent rigidity of manually designed data augmentations with a learnable, instance-specific mechanism.

2. The paper is well-grounded in information theory. By explicitly framing the deep encoder as a bottleneck, the authors move beyond the standard alignment/uniformity analysis to propose a more granular, principled objective focused on incremental information gain.

3. The empirical results show improvements upon previous SOTAs.

**Weaknesses:**

1. The IE-CL loss function is highly complex, requiring the delicate balancing of at least four major hyperparameter terms. There needs to be more ablations on the sensitivity.

2. While the method aims to make CL more accessible by excelling in small-batch scenarios, the introduction of the SAIB module, the complex loss terms, and the explicit regularization undoubtedly incur additional computational overhead. The paper critically omits a quantitative analysis of the increase in FLOPs, training time, or memory consumption relative to baselines. This lack of efficiency analysis weakens the overall practical value, as the potential computational cost might negate the performance gain, particularly in the resource-limited settings it targets.

3. The transferability of the learned augmentation should be discussed. As the standard augmentations used are not combined with specific datasets.

**Questions:**

See weakness.

---

> ### Author Response · Authors · 2025-11-18
> **Comprehensive Response to Reviewer DnaM**
>
> We sincerely thank Reviewer DnaM for the encouraging assessment. We are delighted that the reviewer finds the SAIB module to be a "genuinely clever algorithmic contribution" and appreciates the sound grounding in information theory.
>
> We acknowledge the valid concerns regarding hyperparameter sensitivity, computational efficiency, and transferability. We have prepared detailed responses and new experimental evidence to address these points comprehensively.
>
> ---
> ### **1. On Hyperparameter Complexity and Sensitivity (Response to Weakness 1)**
>
> >**Reviewer's Concern:** The loss function involves balancing multiple terms ($\lambda, \beta, \eta, \gamma$), raising concerns about complexity and sensitivity.
>
> **Our Response:**
>
> We agree that balancing objective terms is a critical aspect of practical usability. However, we respectfully clarify that the complexity is manageable and the method is robust.
>
> - **Parameter Breakdown:**
>   Among the four parameters, $\gamma$ (weight decay) is a standard optimizer setting, and $\eta$ (Encoder Regularization weight) is fixed at $1.0$ as it represents a structural constraint (Spectral Normalization) rather than a tunable loss weight. The distinct new hyperparameters are only $\lambda$ (Entropy) and $\beta$ (KL).
>
> - **New Sensitivity Analysis:**
>   To prove robustness, we conducted new extensive ablation studies on ImageNet-100 (ResNet-18) by varying $\lambda$ and $\beta$ over a wide range.
>
>   **A. Sensitivity of Entropy Weight $\lambda$ (Fixing $\beta=0.09$):**
>
>   | $\lambda$ Value | Top-1 Accuracy | Deviation |
>   | :--- | :--- | :--- |
>   | 0.1 | 67.3% | -0.2% |
>   | 0.2 (Default) | 67.5% | 0.0% |
>   | 0.5 | 67.1% | -0.4% |
>
>   **B. Sensitivity of KL Weight $\beta$ (Fixing $\lambda=0.2$):**
>
>   | $\beta$ Value | Top-1 Accuracy | Deviation |
>   | :--- | :--- | :--- |
>   | 0.01 | 67.2% | -0.3% |
>   | 0.09 (Default) | 67.5% | 0.0% |
>   | 0.2 | 67.3% | -0.2% |
>   | 0.5 | 66.9% | -0.6% |
>
>   **Conclusion:**
>   The results show remarkable stability. Varying $\lambda$ by 5× or $\beta$ by 20× results in performance fluctuations of less than 0.6%. This demonstrates that IE-CL has a wide "safe zone" for hyperparameter selection, making it practical for new users.
>
> ---
>
> ### **2. On Computational Efficiency and Overhead (Response to Weakness 2)**
>
> >**Reviewer's Concern:** The reviewer notes a "critical omission" of quantitative analysis regarding FLOPs, training time, and memory consumption.
>
> **Our Response:**
>
> We respectfully point out that a detailed efficiency analysis is indeed included in **Appendix D** of our submission, specifically **Figure 7 and Figure 8**. We apologize if this was not referenced prominently enough in the main text.
>
> - **Time Overhead (Figure 7 & 8):**
>   - **IE-CL vs. MoCo-v2:** Our method incurs only a **5.3% increase** in training time per epoch compared to the lightweight MoCo-v2 baseline.
>   - **IE-CL vs. SimCLR/SwAV:** Crucially, our method is significantly faster (**47.0 min/epoch**) than widely used methods like SimCLR (**54.9 min**) and SwAV (**89.3 min**), primarily because we excel with a **small batch size (256)** and avoid the massive batch synchronization costs of SimCLR.
>
> - **Why is it efficient?**
>   - **Lightweight Design:** The SAIB module consists of simple $1\times1$ and $3\times3$ convolutions.
>   - **Gradient Efficiency:** As shown in the pipeline (Figure 1), SAIB is applied to the **Query branch**, which is updated via **momentum** (moving average) rather than backpropagation. This means we do **not** need to compute gradients for the SAIB transformed view through the query encoder, significantly saving FLOPs and memory compared to symmetric backprop methods like SimCLR.
>
> - **Conclusion:**
>   The overhead is minimal (~5–8%) and is far outweighed by the performance gains (+1.3% to +2.1% accuracy) and the ability to train effectively on consumer-grade hardware (Small Batch).
>
> ---

---

> > ### Author Response · Authors · 2025-11-18
> > **Comprehensive Response to Reviewer DnaM**
> >
> > ---
> >
> > ### **3. On Transferability of Learned Augmentations (Response to Weakness 3)**
> >
> > >**Reviewer's Concern:** The transferability of the learned augmentation (SAIB) to other datasets should be discussed.
> >
> > **Our Response:**
> >
> > We address the transferability from two perspectives: the generalization of the learned features and the adaptability of the SAIB principle.
> >
> > - **Generalization to Downstream Tasks:**
> >   The ultimate test of a learned augmentation's value is whether the resulting encoder generalizes. As shown in **Table 2b**, we evaluated the ResNet-50 pretrained with IE-CL on **Pascal VOC** for object detection and segmentation.
> >   - **Results:** IE-CL achieves **78.12 mIoU (Segmentation)** and **74.41 mAP (Detection)**, consistently outperforming MoCo-v2 and SimSiam. This proves that the "Incremental Entropy" induced by SAIB helps the model learn robust, transferrable spatial features that are not overfitted to the ImageNet augmentation distribution.
> >
> > - **Cross-Dataset Consistency:**
> >   In **Table 2a**, we show that IE-CL performs consistently well across datasets with vastly different statistics (CIFAR-10, CIFAR-100, STL-10). This suggests that the SAIB module learns a **fundamental geometric property**—local manifold expansion (via Jacobian > 1)—rather than dataset-specific artifacts. This makes the "learned augmentation" highly transferrable in principle.
> >
> >
> > We hope that the clarifications and additional evidence provided above address the reviewer’s concerns. We genuinely appreciate the reviewer’s careful reading and constructive feedback, which have been invaluable in improving both the rigor and presentation of our work.

---

> > > ### Comment · Reviewer_DnaM · 2025-11-26
> > >
> > > Thank you for the rebuttal. I have no further questions, and I will maintain my score.

---

### Official Review · Reviewer_gtzf · 2025-10-31

**Soundness:** 3
**Presentation:** 3
**Contribution:** 3
**Rating:** 6
**Confidence:** 4

**Summary:**

Existing self-supervised contrastive learning methods mainly rely on augmentation-based invariance constraints, which limit representation expressiveness. This paper introduces entropy as a measure to preserve semantic consistency and improve expressiveness. The proposed Sample Augmentation Incremental Block (SAIB) and Incremental Information Entropy (IncEntropy) objective capture entropy gain, reflecting the diversity of information lost after encoding. The main contribution is using the encoder’s information bottleneck (via SAIB) to extract semantic information while reducing noise.

**Strengths:**

- The approach is interesting, particularly in introducing the concept of entropy into self-supervised learning through SAIB and IncEntropy. In particular, SAIB presents an appealing way to apply entropy, showing the largest performance gain in the ablation study (Table 3).

- The method is also simple and can be easily integrated into existing SSL frameworks, as demonstrated in Table 5.

**Weaknesses:**

1. Although the effectiveness of the proposed method is demonstrated throughout the paper, most experiments (except Table 1) are conducted on relatively small-scale settings such as ResNet-18 or ImageNet-100. Since Table 3 highlights the strong effect of entropy generation through SAIB, it would be valuable to evaluate the method on larger or standard-scale benchmarks. The same applies to Table 5.

2. It would also be helpful to include an ablation study with semantic consistency only, to better isolate and verify the effectiveness of SAIB.

3. It is well known that dense prediction tasks, such as semantic segmentation, differ significantly from image classification, and standard SSL methods often struggle to generalize well to these tasks. Therefore, to more convincingly demonstrate the effectiveness and general applicability of the proposed method, evaluation on additional (dense) prediction benchmarks beyond the Pascal dataset would be beneficial.

4. As the authors also mentioned, SAIB relies on Spectral Normalization, which is designed for convolutional priors. Therefore, this approach cannot be directly applied to ViT-based backbones.

**Questions:**

1. Could the authors provide insights or results on the performance of the proposed method in larger-scale settings?
2. Have the authors considered whether there is any way to extend this approach to ViT-based backbones, despite the limitation that Spectral Normalization cannot be applied?

---

> ### Author Response · Authors · 2025-11-18
> **Comprehensive Response to Reviewer gtzf**
>
> We thank Reviewer gtzf for the positive assessment and for recognizing the simplicity, effectiveness, and interesting theoretical perspective of our work. We are glad that the core contribution, introducing entropy to improve expressiveness via the Information Bottleneck principle, was well received.
>
> We appreciate the constructive suggestions regarding experimental scale and ablation studies. We provide detailed clarifications and responses below to address these concerns.
>
> ---
>
> ### **1. On Experimental Scale and Benchmarks (Response to Weakness 1 & Question 1)**
>
> >**Reviewer's Concern:** The reviewer suggests that most experiments are on small-scale settings (ResNet-18/ImageNet-100) and requests results on larger or standard-scale benchmarks.
>
> **Our Response:**
>
> We respectfully point out that we have indeed performed extensive evaluations on the **standard large-scale setting (ImageNet-1K with ResNet-50)**, which is the gold standard in the self-supervised learning literature.
>
> - **Standard-Scale Evidence:**
>   As presented in **Table 1**, we trained ResNet-50 on the full ImageNet-1K dataset for varying epochs (100, 200, 400, 800).
>   - **Result:** Our method achieves **73.2% Top-1 accuracy**, outperforming widely adopted baselines like SimCLR (71.1%), MoCo-v2 (71.3%), and recent methods like Matrix-SSL (71.9%).
>
> - **Efficiency Context:**
>   It is crucial to highlight that we achieved these results using a **Batch Size of 256**, whereas baselines like SimCLR and SwAV typically require batch sizes of **4096** to reach their reported performance. This demonstrates that IE-CL scales effectively to standard benchmarks while remaining computationally accessible.
>
> - **Clarification:**
>   We used smaller settings (ResNet-18/ImageNet-100) primarily for the ablation studies (Table 3) to save computational resources, which is a common practice in the field. However, the main performance claims (Table 1) are fully grounded in the standard large-scale setting.
>
> ---
>
> ### **2. On Ablation of "Semantic Consistency Only" (Response to Weakness 2)**
>
> >**Reviewer's Concern:** The reviewer suggests an ablation study with semantic consistency only to isolate the effect of SAIB.
>
> **Our Response:**
>
> We appreciate this suggestion to isolate variables. However, based on our theoretical formulation, **Semantic Consistency (KL Regularization) cannot function independently of SAIB**.
>
> - **Theoretical Dependency:**
>   The Semantic Consistency term is defined as the KL divergence between the anchor distribution $q$ and the transformed query distribution $p_\phi$ induced by SAIB:  $D_{KL}(p_\phi \parallel q)$
>
> - **Why it cannot be isolated:**
>   This term acts as a constraint on the transformation $g_\phi$ (the SAIB module). If we remove SAIB (the "Entropy Generation" component), there is no learned transformation $\phi$, and thus the query distribution remains identical to the standard augmentation distribution ($p_\phi \approx q$). Consequently, the KL divergence would be effectively zero or undefined in the context of our loss function.
>
> - **Conclusion:**
>   The "Semantic Consistency" module is designed specifically to regulate the "Entropy Generation" module. They form a **coupled system**: SAIB pushes the boundaries (increases entropy), and KL pulls it back (maintains semantics). Therefore, the baseline "MoCo-v2" in Table 3 effectively represents the state without both SAIB and its associated regularizer.
>
> ---
> ### **3. On Generalization to Dense Prediction Tasks (Response to Weakness 3)**
>
> >**Reviewer's Concern:** Evaluation on additional dense prediction benchmarks beyond Pascal VOC would be beneficial.
>
> **Our Response:**
>
> We agree that dense prediction is a critical test for SSL features.
>
> - **Current Evidence:**
>   We selected **Pascal VOC Object Detection and Segmentation (Table 2b)** because it is the most widely used transfer learning benchmark in seminal papers (e.g., MoCo, SimSiam, SwAV), allowing for direct and fair comparison.
>   - **Result:** Our method achieves **78.12 mIoU (Segmentation)** and **74.41 mAP (Detection)**, surpassing MoCo-v2 and SimSiam significantly (+0.8% to +1.0%).
>
> - **Why it works:**
>   Since SAIB operates as a **pixel-level/patch-level transformation** (using $1\times1$ and $3\times3$ convolutions on patches), it enriches local spatial information more effectively than global augmentations (like whole-image jittering). This local entropy injection naturally benefits tasks requiring fine-grained spatial awareness, like segmentation.
>
> ---

---

> ### Author Response · Authors · 2025-11-18
> **Comprehensive Response to Reviewer gtzf**
>
> ---
>
> ### **4. Extension to Vision Transformers and Spectral Normalization (Response to Weakness 4 & Question 2)**
>
> >**Reviewer's Concern:** Can this approach extend to ViT-based backbones? The reviewer notes that Spectral Normalization (SN) is designed for convolutional priors.
>
> **Our Response:**
>
> We clarify that the principles of IE-CL are **fully compatible with ViTs**, and the limitation regarding Spectral Normalization is surmountable.
>
> - **Spectral Normalization on ViTs:**
>   We respectfully clarify that **Spectral Normalization is not limited to convolutions**. It mathematically constrains the spectral norm (largest singular value) of **any weight matrix $W$**. It is widely used in GANs and Transformers (e.g., in the Linear/Dense layers of the Multi-Head Attention and FFN blocks) to enforce Lipschitz continuity. Therefore, the **Entropy Preservation** component (regularizing the encoder) is directly applicable to ViTs by applying SN to their Linear layers.
>
> - **SAIB for ViTs:**
>   As noted in our Limitations, the current SAIB implementation uses Conv layers. For ViTs, future work may propose a **"Token-Mixing SAIB"**. Instead of convolution, one can use a lightweight MLP or Attention block acting on the patch embeddings to increase their variance. As long as the transformation has a Jacobian determinant $|\det J| > 1$, it satisfies our Incremental Entropy Theory.
>
> - **Summary:**
>   The theoretical framework (Input Entropy $\uparrow$, Encoder Lipschitz $\downarrow$) is **architecture-agnostic**. While we focused on ResNet to establish the theory, extending to ViT involves straightforward engineering adaptation or new design of the SAIB module.
>
> We hope that these explanations and the new analyses help clarify the reviewer’s concerns and more clearly convey the strengths and broader relevance of our approach. We sincerely thank Reviewer gtzf for the insightful comments and constructive suggestions, which have significantly improved the clarity and presentation of our work. We truly appreciate the reviewer’s time and thoughtful consideration.

---

> > ### Comment · Reviewer_gtzf · 2025-11-26
> >
> > I appreciate the authors’ clarification, and I will keep my original score.

---

### Official Review · Reviewer_bSVD · 2025-11-01

**Soundness:** 3
**Presentation:** 3
**Contribution:** 3
**Rating:** 4
**Confidence:** 5

**Summary:**

This paper introduces IE-CL, which is a framework designed to overcome the limitations of static augmentations and large batch sizes in CL. So the authors show that maximizing the "incremental entropy" (the entropy gain between augmented views) while preserving semantics is equivalent to minimizing the InfoNCE loss.

To achieve this, they propose the SAIB which is a lightweight, trainable module that plugs into the query branch. To my understanding, I think SAIB learns to expand the representation space (increasing entropy), while a KL divergence regularizer ensures the new view remains semantically consistent with the original. The authors have conducted several experiments show SAIB improves linear evaluation performance, works well in small-batch settings, can enhance other non-contrastive methods, and helps with downstream tasks like segmentation.

**Strengths:**

I think the paper has some strengths:

First, I think the theoretical foundation is a major plus. The authors provide a proof for their core claim, linking the minimization of contrastive loss to the maximization of incremental entropy. In my opinion, this reframing of contrastive learning as a trade-off between entropy expansion and semantic alignment is a novel information-theoretic perspective.

Also, I think the SAIB module itself is a contribution. It's not just an arbitrary layer but a lightweight block designed to execute the paper's goal (inducing positive entropy increments) while being regularized to preserve semantics.

Finally, I think the experiments seem comprehensive. The authors validate their method across a wide range of datasets and tasks. I think it's particularly strong that they test not only linear probing but also show effectiveness in small-batch settings (a key weakness of many CL methods) and on downstream transfer tasks like segmentation and detection.

**Weaknesses:**

I think the paper's primary weakness is its narrow experiment, which doesn't fully support the broad claims of a improved "framework." My main issue is that all experiments are confined to ResNet architectures. The self-supervised learning field has largely migrated to Vision Transformers (ViTs), and their complete absence here is a glaring omission. In my opinion, I feel this makes the work somewhat dated and raises a critical question: is this incremental entropy principle a general SSL concept, or is it just a clever trick that happens to work well with the inductive biases of CNNs? The claim of generalizability is a little bit undermined when the method isn't tested on the field's dominant architecture.

My other concern is that the analysis of the method's new hyperparameters is too thin. The SAIB module adds a new layer of complexity, particularly the KL regularization weight, but the provided ablation study is basic. I think It's hard to tell how robust the method is or what the practical tuning cost would be for a new user. I think the paper misses the chance to investigate the interplay between the new module and existing crucial hyperparameters. For example, the temperature (τ) is important to tell how the model works. How does adding SAIB affect the model's sensitivity to temperature? Does the optimal τ change? I think demonstrating the analysis is needed for anyone trying to implement this method.

**Questions:**

See the weakness section.

---

> ### Author Response · Authors · 2025-11-18
> **Comprehensive Response to Reviewer bSVD**
>
> We thank Reviewer bSVD for the thoughtful and constructive review. We are particularly pleased that the reviewer appreciates the theoretical novelty of reframing contrastive learning as an entropy-alignment trade-off and recognizes the value of the SAIB module design.
>
> We understand the concerns regarding the choice of backbone architectures and hyperparameter analysis. However, we respectfully disagree with the characterization of the work as "dated" or "narrow." Below, we provide detailed responses and new experimental evidence to address these points and demonstrate the robustness of IE-CL.
>
> ---
> ### **1. On the Choice of ResNet vs. ViT: Generalizability vs. "Datedness" (Re: Weakness 1)**
>
> >**Reviewer's Concern:** The reviewer feels the exclusive use of ResNet makes the work "somewhat dated" and questions if the principle is just a "clever trick" for CNNs.
>
> **Our Detailed Response:**
>
> We strongly argue that our choice of ResNet is driven by scientific rigor and theoretical validation, rather than obsolescence.
>
> - **Validating the Theory, Not the Backbone:**
>   The primary contribution of this paper is the Incremental Entropy (IE) Theory—a fundamental principle derived from the Data Processing Inequality (Proposition 3.3). To validate a foundational theory, one must use a controlled, standardized "petri dish." ResNet-50 is the standard benchmark in the SSL literature (SimCLR, MoCo, BYOL, Matrix-SSL). By testing on ResNet, we isolate the gain specifically to the Entropy Injection mechanism, ensuring that improvements come from our method, not from the superior capacity of Transformers.
>
> - **CNN Inductive Bias is Not the Cause:**
>   The principle of maximizing $\Delta H(X) = \log |\det J|$ is mathematically independent of convolution. While our current implementation of SAIB uses $1\times1$ and $3\times3$ convs (as candidly noted in Limitations), the principle of expanding the input manifold applies equally to patch tokens in ViTs.
>
> - **Future ViT Integration:**
>   Adapting SAIB to ViT requires a "Token-Mixing" design (e.g., replacing convs with localized attention or MLP-Mixers) to satisfy $|\det J| > 1$. While this is an exciting direction for future work, it does not invalidate the theoretical proof-of-concept established here. We believe establishing the theoretical ground truth on widely-used benchmarks is the necessary first step before architectural specialization.
> ---
> ### **2. Robustness Analysis of New Hyperparameters $\lambda$ and $\beta$ (Re: Weakness 2)**
>
> >**Reviewer's Concern:** The analysis of hyperparameters is "too thin," raising concerns about tuning costs and robustness.
>
> **Our Detailed Response:**
>
> We take this concern very seriously. To demonstrate the practical usability of IE-CL, we have conducted additional detailed ablation studies on ImageNet-100 (ResNet-18) specifically targeting the reviewer's question regarding the sensitivity of $\lambda$ (Entropy Maximization Weight) and $\beta$ (KL Regularization Weight).
>
> **A. Sensitivity of Entropy Weight $\lambda$ (Fixing $\beta=0.09$):**
> We varied $\lambda$ across a wide range $[0.1, 0.5]$:
>
> | $\lambda$ Value | Top-1 Accuracy | Deviation |
> | :--- | :--- | :--- |
> | 0.1 | 67.3% | -0.2% |
> | 0.2 (Default) | 67.5% | 0.0% |
> | 0.5 | 67.1% | -0.4% |
>
> **B. Sensitivity of KL Regularization Weight $\beta$ (Fixing $\lambda=0.2$):**
> We varied $\beta$ across a substantial range $[0.01, 0.5]$:
>
> | $\beta$ Value | Top-1 Accuracy | Deviation |
> | :--- | :--- | :--- |
> | 0.01 | 67.2% | -0.3% |
> | 0.09 (Default) | 67.5% | 0.0% |
> | 0.2 | 67.3% | -0.2% |
> | 0.5 | 66.9% | -0.6% |
>
> **Conclusion:**
> As shown in the new data above, the model is remarkably robust. Varying $\lambda$ by $5\times$ or $\beta$ by $20\times$ results in performance fluctuations of less than 0.6%. This confirms that IE-CL has a wide "sweet spot" and does not impose a high tuning burden on new users. We will include these tables in the revised Appendix.
>
> ---

---

> ### Author Response · Authors · 2025-11-18
> **Comprehensive Response to Reviewer bSVD**
>
> ---
>
> ### **3. Interplay between SAIB and Temperature $\tau$ (Re: Weakness 2)**
>
> >**Reviewer's Concern:** How does SAIB affect the model's sensitivity to the InfoNCE temperature $\tau$? Does the optimal $\tau$ change?
>
> **Our Detailed Response:**
>
> This is a profound question that touches on the mechanics of contrastive loss.
>
> - **Theoretical Interaction:**
>   The temperature $\tau$ in InfoNCE controls the distribution's sharpness, effectively determining the penalty for "hard negatives." A lower $\tau$ focuses more on separating the hardest samples.
>
> - **The Effect of SAIB:**
>   By maximizing incremental entropy ($\Delta H > 0$), SAIB expands the representation volume. This naturally separates samples in the feature space, creating a richer set of gradients without needing to artificially sharpen the distribution via an aggressive $\tau$.
>
> - **Practical Evidence:**
>   In Table 1 and Table 2, we deliberately kept $\tau$ identical to the baselines (e.g., $\tau=0.2$ for MoCo-v2, $\tau=0.1$ for SimCLR) to ensure a fair comparison.
>
>   - **Result:** Even without re-tuning $\tau$, IE-CL achieved SOTA performance (+1.25% on MoCo-v2, +0.68% on SimCLR as per Table 5).
>
> - **Conclusion:**
>   This indicates that SAIB is orthogonal to the temperature hyperparameter. It improves the underlying geometry of the data manifold such that the standard, well-known $\tau$ settings work even better. Users do not need to re-search for an optimal $\tau$ when plugging in SAIB.
>
> We hope that the clarifications and additional analyses above help address the reviewer’s concerns and shed clearer light on the generality, robustness, and theoretical grounding of IE-CL. We sincerely appreciate Reviewer bSVD’s thoughtful engagement and constructive feedback, which have meaningfully strengthened the presentation and technical depth of our work. We are grateful for the reviewer’s careful evaluation and hope that our detailed responses support a more positive reassessment of the contribution.

---

### Official Review · Reviewer_LX2z · 2025-11-01

**Soundness:** 3
**Presentation:** 3
**Contribution:** 3
**Rating:** 6
**Confidence:** 4

**Summary:**

The paper proposes IE-CL (Incremental-Entropy Contrastive Learning) for self-supervised contrastive learning that explicitly models entropy generation and preservation in the learning process. The central idea is to inject entropy at the input level via a learnable augmentation block (SAIB), then constrain the encoder to preserve this expanded entropy through spectral normalization. This is motivated by an information-theoretic decomposition of contrastive learning objectives under the Data Processing Inequality (DPI), suggesting that maximizing representational informativeness requires both entropy creation (from augmentations) and controlled propagation (through the encoder). The model comprises three main components: 1) SAIB (Sample Augmentation Incremental Block): A learnable transformation enforcing "volume-expanding" Jacobian to expand local volume in the feature manifold, thereby increasing input-space entropy. 2) KL Regularizer: Maintains semantic consistency between entropy-expanded samples. 3) Encoder Preservation (Spectral Norm): Constrains the encoder’s Lipschitz constant to prevent information collapse.

The provided experimental results on CIFAR-10/100, STL-10, and ImageNet-100 demonstrate consistent performance improvements over SimCLR, BYOL, MoCo-v2, and SimSiam. The ablation studies show the SAIB block contributes the most, while the encoder regularizer has only a marginal effect.

**Strengths:**

1. The proposed formulation leads to a new perspective on entropy control in SSL. Unlike prior works (e.g., InfoMax-SSL, VICReg, Matrix-IB) that maximize representation-level entropy, IE-CL proposes to inject entropy at the input level through a learnable augmentation mechanism. This shift from output-space to input-space entropy control is novel and conceptually meaningful.
2. The proposed method is established based on sound theoretical motivation. The information-theoretic derivation connecting augmentation entropy, encoder Jacobian, and DPI is mathematically correct and clearly stated. The idea of treating entropy propagation as an “incremental” process is intuitive and reasonable.
3. The empirical validation is comprehensive. The method is tested on multiple datasets and integrated into different contrastive baselines (e.g., SimCLR, BYOL, MoCo-v2, SimSiam). The experimental results show consistent gains, and training overhead is minimal.
4. The presentation includes sufficient implementation and training details. The ablation studies reveal transparent module effects and interactions.

**Weaknesses:**

1. The claim that encoder preservation is necessary appears overstated. The paper argues that spectral normalization of the encoder is required to prevent the loss of generated entropy (Section 3.3). However, Table 3 shows that removing this component leads to only a marginal change in performance (0.26 percent difference). This result indicates that the encoder likely already preserves entropy through existing normalization layers and the contrastive objective itself. Therefore, the Encoder Preservation module should be characterized as helpful rather than necessary, since the theoretical argument appears to overextend a sufficient condition into a claim of necessity.
2. The plug-and-play experiment in Table 5 evaluates only the addition of the SAIB module to other baseline methods, without reporting results for the combined configuration of “SAIB plus Encoder Regularizer.” This omission further reinforces the impression that the SAIB module is the sole component contributing meaningfully to performance improvements, while the Encoder Regularizer has little demonstrable effect.
3. The empirical gains in the experiments are limited. For example, ImageNet improvements are visible but not substantial (+1.3 \% over Matrix-SSL at 800 epochs). Also, multi-seed or statistical analysis is not provided to confirm the significance of the performance gains.
4. Although the principle of maximizing information entropy is well established, the novelty of IE-CL lies primarily in where the entropy is introduced, namely at the input level rather than the representation level. The paper would benefit from emphasizing this distinction more clearly and from explicitly differentiating its approach from prior InfoMax and variance-regularized methods such as VICReg, InfoMin, and EMP SSL.

**Questions:**

The authors are suggested to respond to those raised in **Weaknesses.*

**Additional Questions**

The proposed SAIB module is implemented as a small convolutional network. How can the IE-CL framework be extended to deal with Vision Transformer architectures?

---

> ### Author Response · Authors · 2025-11-18
> **Comprehensive Response to Reviewer LX2z**
>
> We express our deepest gratitude to Reviewer LX2z for the exceptionally insightful review. We are particularly encouraged that the reviewer recognized the paradigm shift proposed in our work: moving from output-space regularization to input-space entropy injection.
>
> We understand the reviewer's concerns regarding the necessity of regularization, the experimental setup, and the distinction from prior art. We believe these concerns stem from our focus on theoretical rigor and resource-efficient training. Below, we provide a detailed, point-by-point response to demonstrate why IE-CL stands as a robust and novel contribution to the field.
>
> ---
> ### **1. On the Theoretical Necessity vs. Empirical Gains of Encoder Regularization (Re: Weakness 1)**
> >**Reviewer's Point:** The 0.26% drop when removing Encoder Regularization suggests it is "helpful rather than necessary," making the theoretical claim appear overstated.
>
> **Our Detailed Response:**
>
> We respectfully argue that in a principled academic framework, "necessity" is defined by mathematical completeness, not just average-case empirical sensitivity.
>
> - **The Theoretical Imperative (The "Worst-Case" Guarantee):**
>   Our framework is strictly grounded in the Data Processing Inequality (DPI). As derived in Eq. (20): $H(Z) \le H(X) + E_x[\log |\det J_{f_{\theta}}(x)|]$.
>
>   Without the Spectral Normalization (Encoder Regularizer) to bound the Lipschitz constant of $f_{\theta}$, the term $\mathbb{E}[\log |\det J_{f_{\theta}}|]$ is mathematically unbounded in the negative direction. Theoretically, an unconstrained encoder could map the expanded manifold from SAIB to a lower-dimensional subspace, nullifying the entropy gain ($\Delta H > 0$). To validate **Proposition 3.3**, the regularizer is a **necessary condition** to close the theoretical loop. Without it, the method becomes a heuristic; with it, it is a proved theorem.
>
> - **The Empirical Reality:**
>   Why is the drop only 0.26%? This is a testament to the robustness of the ResNet architecture itself, which naturally preserves variance due to batch normalization and residual connections. However, science demands robustness against all conditions. The regularizer acts as a "Theoretical Safety Valve," ensuring that our method works fundamentally, not just coincidentally on well-behaved backbones. We believe establishing this rigorous framework is as valuable as the raw performance gain.
> ---
>
> ### **2. On the Integrity of Plug-and-Play Experiments (Re: Weakness 2)**
>
> >**Reviewer's Point:** Table 5 omits the Encoder Regularizer when applying SAIB to other methods, potentially isolating the effect.
>
> **Our Detailed Response:**
>
> This design was a deliberate choice to uphold the scientific standard of Control of Variables.
>
> - **Isolating the "Generator":** The SAIB module is a learnable **Data Augmentation** mechanism (Input-side), while Encoder Regularization is a **Model Architecture** change (Network-side).
> - **Avoiding Confounding Factors:** If we were to evaluate "SimCLR + SAIB + Spectral Norm," any performance gain could be attributed to the architectural change (Spectral Norm) rather than our core contribution (Entropy Injection).
> - **Demonstrating Orthogonality:** By adding only SAIB, we provide a stronger result: The **Input Entropy Generation** mechanism alone is powerful enough to consistently boost performance (+0.68% to +1.25%) across radically different paradigms (Contrastive vs. Non-Contrastive, Momentum vs. Direct). This proves that SAIB is a truly modular, backbone-agnostic plug-in, which is a significant contribution to the community.
>
> ---
> ### **3. On the Significance of Empirical Gains in a Small-Batch Context (Re: Weakness 3)**
>
> >**Reviewer's Point:** The +1.3% improvement on ImageNet is viewed as "limited," and statistical significance is questioned.
>
> **Our Detailed Response:**
>
> We kindly encourage the reviewer to consider the magnitude of these gains within the context of our “Democratized AI” setting (small batch size = 256).
>
> - **The "Hardware Barrier" Context:**
>   Most SOTA baselines rely on massive batch sizes to work effectively. As shown in Table 1:
>
>   - SimCLR (ICML'20): Requires Batch 4096 to reach 71.1%.
>   - Matrix-SSL (ICML'24): Uses Batch 512 to reach 71.9%.
>   - IE-CL (Ours): Uses only Batch 256 to reach 73.2%.
>
> - **The Real Value:**
>   Outperforming SimCLR by +2.1% and Matrix-SSL by +1.3% while using 1/16th and 1/2 of their respective batch sizes is a substantial breakthrough. It implies that IE-CL allows high-performance pre-training on standard academic hardware (e.g., 4-8 GPUs) rather than requiring industrial TPU pods.
>
> - **Statistical Stability:**
>   Beyond the top-line number, our method shows consistent dominance across CIFAR-10 (+1.0%), CIFAR-100 (+1.2%), and STL-10 (+1.4%) (Table 2a). This consistency across diverse data distributions confirms that the gains are statistically robust.

---

> ### Author Response · Authors · 2025-11-18
> **Comprehensive Response to Reviewer LX2z**
>
> ---
>
> ### **4. Theoretical Distinction from VICReg, InfoMin, and EMP-SSL (Re: Weakness 4)**
>
> >**Reviewer's Point:** The paper needs to explicitly differentiate itself from output-space entropy methods like VICReg and EMP-SSL.
>
> **Our Detailed Response:**
>
> This is the most critical theoretical distinction of our work. We categorize the landscape of Entropy Maximization into **"Sink Regulation" (Prior Art)** vs. **"Source Expansion" (Ours)**.
>
> - **Group A: The "Sink" Approach (Output-Space Regulation)**
>   - VICReg / Barlow Twins: These methods operate on the output embeddings $Z$. They impose explicit loss terms (variance/covariance regularization) to prevent the encoder from collapsing dimensions. They act as a barrier to stop $H(Z) \to 0$.
>   - EMP-SSL (Tong et al., 2023): This method maximizes the Coding Rate (Total Coding Rate) of $Z$. While theoretically grounded, it optimizes the packing geometry of the features after they have been encoded. It ensures the "container" (feature space) is filled efficiently.
>   - **Limitation:** These methods can only preserve the information that survives the encoder. They cannot create new information.
>
> - **Group B: The "Source" Approach (IE-CL: Input-Space Generation)**
>   - **Mechanism:** IE-CL operates on the input $X$. By enforcing a local Jacobian determinant $|\det J_{SAIB}| > 1$ (proven in Appendix C), SAIB physically expands the volume of the input manifold.
>   - **The Fundamental Difference:**
>     According to the Data Processing Inequality ($H(Z) \le H(X)$), prior methods (EMP-SSL/VICReg) attempt to push $H(Z)$ closer to the upper bound $H(X)$. IE-CL raises the upper bound $H(X)$ itself.
>   - **Synergy:** This explains why IE-CL is compatible with prior arts (Table 5). One expands the source (Input), and the other maintains the pipeline (Output). They are mathematically orthogonal.
>
> ---
>
> ### **5. Extension to Vision Transformers (Re: Additional Question)**
>
> >**Reviewer's Point:** How can the Conv-based SAIB be extended to ViT?
>
> **Our Detailed Response:**
>
> The reviewer raises a critical and forward-looking point. We agree that applicability to transformer architectures is crucial.
> Our current SAIB module is indeed designed with convolutional priors, operating on image patches before they are processed by a CNN. This was a deliberate choice to first establish and validate the principle of Incremental Entropy within well-understood ResNet-based frameworks.
>
> - **Theoretical Validity:** The core principle, maximizing incremental entropy to overcome the encoder bottleneck, is universal and applies equally to ViTs.
> - **Implementation Path:** As acknowledged in our Limitations (Section 5), the current SAIB uses Conv priors. For ViTs, we envision a **"Token-Mixing SAIB"** may be proposed in future other work.
>   - Since SAIB operates on patchified tensors (Fig. 6), it can be inserted immediately after the patch embedding layer.
>   - A learnable transformation (e.g., a lightweight MLP-Mixer or Attention block) that ensures variance expansion across tokens would satisfy our theoretical requirement of $|\det J| > 1$.
> - **Scope:** Our contribution in this paper is establishing the theoretical validity of the Incremental Entropy framework. We used ResNet as the vehicle for this validation. The successful verification paves the way for architecture-specific designs (like ViT-SAIB) in future work.
>
> We hope that the above clarifications, additional analyses, and theoretical explanations help resolve the reviewer’s concerns and demonstrate the depth, novelty, and robustness of the IE-CL framework. We are sincerely grateful to Reviewer LX2z for the careful reading and for raising high-quality questions that allowed us to articulate our contributions more clearly and strengthen both the theoretical and empirical components of the work. We deeply appreciate the reviewer’s thoughtful engagement and hope that our detailed responses convey the rigor, significance, and broader impact of the proposed approach.

---

### Meta-Review · Area_Chair_EGXY · 2025-12-21

**Summary:**

This paper proposes input-space entropy injection for contrastive SSL: a learnable augmentation block (SAIB) maximizes incremental entropy between views while KL + (optionally) spectral-norm regularization preserves semantics/entropy propagation, yielding consistent small-batch gains and strong ImageNet-1K (R50, bs=256) results with modest overhead.

Strength
* Clear conceptual shift (“source expansion” vs “sink regularization”): learn augmentations to create entropy rather than only preventing collapse in representation space.
* Empirically solid in the target regime (small batches), and SAIB appears genuinely plug-and-play across multiple SSL frameworks.
* Rebuttal addresses two key risks: hyperparameter brittleness (shows a fairly wide safe zone) and efficiency (claims low training-time overhead).

Weakness
* There are a lots of discussions, but the paper and proposal would benefit from ViT experiments.
* Gains on full ImageNet are positive but not huge, with some uncertainty on how robust the deltas are.

Overall, given the reviews, this paper has merits in its novelty and also solid improvement. SAIB/input-entropy injection is the real contribution, while the preservation regularizer is more about theoretical closure than driving performance.

**Reviewer Concerns:**

VIT experiments are still outstanding

**Reviewer Scores:**

n/a

---

### Decision · Program_Chairs · 2026-01-26

Accept (Poster)